# Cyclin A2 degradation during the spindle assembly checkpoint requires multiple binding modes to the APC/C

Suyang Zhang[1,2], Thomas Tischer [iD] [1] & David Barford [iD] [1]

The anaphase-promoting complex/cyclosome (APC/C) orchestrates cell cycle progression by controlling the temporal degradation of specific cell cycle regulators. Although cyclin A2 and cyclin B1 are both targeted for degradation by the APC/C, during the spindle assembly checkpoint (SAC), the mitotic checkpoint complex (MCC) represses APC/C's activity towards cyclin B1, but not cyclin A2. Through structural, biochemical and in vivo analysis, we identify a non-canonical D box (D2) that is critical for cyclin A2 ubiquitination in vitro and degradation in vivo. During the SAC, cyclin A2 is ubiquitinated by the repressed APC/C-MCC, mediated by the cooperative engagement of its KEN and D2 boxes, ABBA motif, and the cofactor Cks. Once the SAC is satisfied, cyclin A2 binds APC/C-Cdc20 through two mutually exclusive binding modes, resulting in differential ubiquitination efficiency. Our findings reveal that a single substrate can engage an E3 ligase through multiple binding modes, affecting its degradation timing and efficiency.

[1] MRC Laboratory of Molecular Biology, Francis Crick Avenue, Cambridge CB2 0QH, UK. [2] Present address: Max Planck Institute for Biophysical Chemistry, Göttingen 37077, Germany. Correspondence and requests for materials should be addressed to D.B. (email: dbarford@mrc-lmb.cam.ac.uk)

Controlled cell-cycle progression is governed by the interplay of ordered protein degradation and reversible protein phosphorylation. In eukaryotes, the anaphase-promoting complex/cyclosome (APC/C), through targeting specific cell-cycle regulators for degradation, regulates chromatid segregation at the metaphase to anaphase transition, the exit from mitosis and the establishment and maintenance of G1[1–3]. APC/C-substrate selection is controlled by two coactivator subunits, Cdc20 and Cdh1, that recognize three major classes of degrons: the destruction box (D box)[4], KEN box[5] and ABBA motif[6–9]. In addition, coactivators stimulate APC/C catalytic activity[10–12].

The spindle assembly checkpoint (SAC) ensures the equal distribution of chromosomes to the two daughter cells by arresting the cell in metaphase until all chromosomes are correctly attached to kinetochores[13,14]. In prometaphase, the APC/C targets cyclin A2 for degradation, whereas its activity towards securin and cyclin B1 is suppressed by the SAC[13,14]. Early degradation of cyclin A2, preceding anaphase onset, is important as a persistent level of cyclin A2 prevents the stable kinetochore-microtubule attachments necessary for chromosome segregation[15]. Consequently, increased levels of cyclin A2 in mammalian cells delayed metaphase and anaphase onset[16,17], and are associated with a variety of tumours[18]. Transgenic mice over-expressing cyclin A2 in mammary glands exhibited hyperplasia and nuclear abnormalities[19]. These studies underlie the importance of understanding the mechanisms responsible for targeted cyclin A2 degradation.

Cyclin A2 is degraded in prometaphase about 30 min before cyclin B1[16,17], due to the SAC-imposed inhibition of the APC/C towards cyclin B1. The SAC represses APC/C^Cdc20 activity through its effector, the mitotic checkpoint complex (MCC), a tetrameric-protein complex consisting of Cdc20, BubR1, Bub3 and Mad2[20]. Through its BubR1 subunit, the MCC inhibits APC/C^Cdc20 by both acting as a pseudo-substrate that blocks all six degron-binding sites on both Cdc20 subunits of APC/C^MCC (refs. [21–23]), and obstructing the E2 binding site to suppress APC/C's catalytic activity[22,23].

Both the N-terminal 165 residues of human cyclin A2 and the Cdk-associated cofactor Cks are necessary and sufficient to allow SAC-resistant degradation of cyclin A2[16,24,25], yet the exact degrons responsible for cyclin A2 degradation are unknown. Unlike cyclin B1, deletion of the canonical D box in cyclin A2 had no influence on either the timing or the rate of its degradation[16,17,26,27], and the D box failed to function as a portable destruction signal[28,29], implying that the canonical D box of cyclin A2 may be dispensable. Here, we identify a previously uncharacterized non-canonical D box (D2 box) in cyclin A2. We define the roles of individual degrons in cyclin A2 and how they cooperate to overcome repression of cyclin A2 ubiquitination by the MCC. Our data indicate that the MCC-repressed state of the APC/C (APC/C^MCC) is responsible for ubiquitinating cyclin A2 during an active SAC by directly engaging cyclin A2. Furthermore, we reveal two distinct binding modes of cyclin A2 to APC/C^Cdc20 that influence its ubiquitination efficiency.

## Results

### Known degrons of cyclin A2 are dispensable for APC/C^Cdc20 activity.
The cell-cycle dependent ordering of substrate degradation is governed by different states of APC/C activity, degron affinity and substrate processivity as well as competition among substrates and modification of degrons[7,30–36]. Because Cdk2-cyclinA2-Cks2 functions both as a kinase and substrate of the APC/C, to understand how cyclin A2 evades the spindle assembly checkpoint, we performed ubiquitination assays in the presence of the Cdk1/2 inhibitor iii. In the absence of this inhibitor, cyclin

A2 ubiquitination is suppressed due to Cdc20 phosphorylation (Supplementary Fig. 1a). In vitro, cyclin A2 and cyclin B1 are substrates of both APC/C^Cdc20 (phosphorylated APC/C) and APC/C^Cdh1, independent of complex formation with Cdk2-Cks2 (Supplementary Fig. 1b, c, e). However, unlike cyclin B1, cyclin A2 was processively ubiquitinated in the presence of the MCC (Supplementary Fig. 1d). Nevertheless, cyclin A2 alone is not sufficient and it requires the associated cofactor Cdk-Cks to overcome the MCC-imposed inhibition of APC/C activity (Supplementary Fig. 1d), consistent with previous studies in vivo[24]. The ability of Cdk2-cyclinA2-Cks2 to overcome the MCC-mediated inhibition of APC/C activity in vitro is dependent on its concentration. At lower concentrations cyclin A2 ubiquitination was inhibited by the MCC, whereas increasing the cyclin A2 concentration allowed it to effectively overcome MCC-imposed inhibition (Supplementary Fig. 1f). The concentrations of the APC/C, MCC and Cdc20 used in the ubiquitination assays are within the range of their reported physiological intracellular concentrations[37,38], whereas those of Cdk2-cyclin substrates are substantially higher than that reported in ref. [39].

In both cyclin A2 and cyclin B1, degrons present within their N-terminal domains target the cyclins for ubiquitination by the APC/C[4,28,40] (Fig. 1a). In addition to the canonical D box (referred as the D1 box), cyclin A2 contains a putative KEN box (DQEN)[41] and an ABBA motif[6] (Fig. 1a). To verify the roles that individual degrons play in targeting cyclin A2 to the APC/C, cyclin A2 mutants were generated in which each degron was mutated individually. Mutation of the putative KEN box (ΔK) alone completely abolished cyclin A2 ubiquitination by APC/C^Cdh1, whereas mutation of the canonical D box (ΔD1) and the ABBA motif (ΔA) did not have an effect (Fig. 1b, c, lanes 1–3 and 5 and Supplementary Fig. 2a). This verified the existence of a KEN box in cyclin A2 and is in agreement with previous findings that the KEN box is important for substrate recognition by APC/C^Cdh1 (ref. [5]). On the other hand, individual mutation of the KEN box (ΔK), the D1 box (ΔD1) and the ABBA motif (ΔA) did not impair cyclin A2 ubiquitination by APC/C^Cdc20 (Fig. 1d, e and Supplementary Fig. 2b). Mutating the KEN box or ABBA motif reduced cyclin A2 ubiquitination by APC/C^MCC, whereas in contrast, disrupting the canonical D1 box had no effect (Fig. 1f, g and Supplementary Fig. 2c). Therefore, mutation of the canonical D1 box did not decrease cyclin A2 ubiquitination in vitro by APC/C^Cdh1, APC/C^Cdc20 or APC/C^MCC, suggesting that the D1 box is dispensable for its ubiquitination. This is consistent with findings that the D1 box is not required for cyclin A2 degradation in cells[24].

To assess whether our in vitro assays reflect the role of cyclin A2 degrons in controlling cyclin A2 degradation in vivo, we introduced cyclin A2 harbouring degron mutations tagged with enhanced GFP into HEK293 FlpIn-TRex cells and monitored their degradation pattern in vivo using live cell microscopy (Fig. 1h–j and Supplementary Movies 1, 2). The expression of cyclin A2 mutant proteins in these cell lines can be finely controlled by addition of doxycycline (Supplementary Fig. 3a) and a concentration was chosen to reflect the expression level of endogenous cyclin A2. While we cannot formally exclude that doubling the protein levels of cyclin A2 in cells by this approach influences the degree of phosphorylation present on Cdk/cyclin substrates, this effect seems to be negligible, because mitotic timing is not altered upon expression of different cyclin A2 mutants (except for non-degradable cyclin A2 mutants, see below) (Supplementary Fig. 3b). Single mutations of individual degrons (KEN box, D1 box or ABBA motif) showed similar destruction timing and kinetics as wild-type cyclin A2 (Fig. 1h, j and Supplementary Fig. 3c, d), consistent with the negligible effect of disrupting these degrons in the in vitro assays for

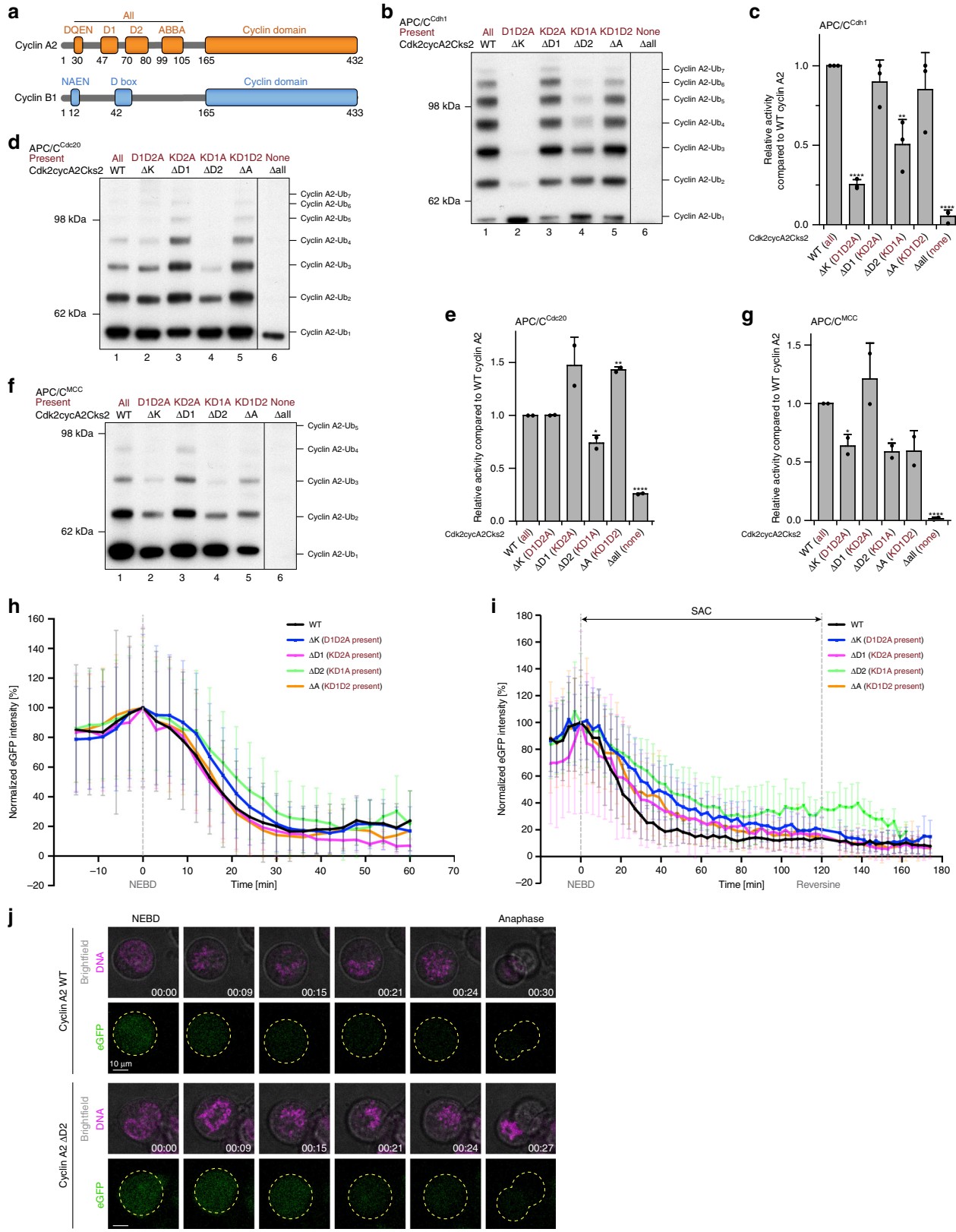

APC/C[Cdc20] and APC/C[MCC]. However, when the HEK cells were arrested with the kinesin-like 11/Eg5 inhibitor S-trityl-L-cysteine (STLC) in prometaphase with an active SAC, all three single degron mutations resulted in a decreased rate of cyclin A2 degradation (Supplementary Fig. 3e), and a slight delay in destruction timing compared to wild-type cyclin A2 (Fig. 1i), implying they may contribute to the SAC-resistant degradation of cyclin A2 in vivo.

To further explore the determinants of cyclin A2 ubiquitination, a mutant was generated with all three known degrons

**Fig. 1** Mutation of individual cyclin A2 degrons did not abolish its ubiquitination and degradation. **a** Domain organization of human cyclin A2 and cyclin B1, with their respective degrons indicated. DQEN and NAEN are the putative KEN boxes. **b–g** Ubiquitination assays of single degron mutations in cyclin A2 by APC/C$^{Cdh1}$ (**b**), APC/C$^{Cdc20}$ (**d**) and APC/C$^{MCC}$ (**f**). In vitro phosphorylated APC/C was used for complex formation with Cdc20 and the MCC, while unphosphorylated APC/C was used for Cdh1. The remaining degrons in cyclin A2 after the mutation are labelled in dark red. Ubiquitination reactions were analysed by Western blotting with an anti-His antibody to detect the ubiquitin-modified substrates. Control gels showing the unmodified substrate for representative reactions are shown in Supplementary Fig. 2. The western-blots were quantified, with error bars indicating standard deviation. The APC/C activity towards cyclin A2 mutants is normalized to ubiquitination of wild-type cyclin A2 and the significance is calculated using unpaired Student's t-test (indicated with stars, $n = 3$ for **c** and 2 for **e** and **g**, Supplementary Table 2). **h, i** Degradation profiles of single degron mutation constructs of eGFP-cyclin A2 in HEK cells during unperturbed mitosis (**h**) and a SAC arrest using STLC (**i**). Under unperturbed mitosis, $n = 62$ wild-type, 49 ΔK, 47 ΔD1, 44 ΔD2 and 37 ΔA cells were analysed, whereas 13 wild-type, 14 ΔK, 15 ΔD1, 22 ΔD2 and 12 ΔA cells were analysed for conditions under a SAC arrest. The timepoints of NEBD and reversine addition to release the SAC arrest are marked, respectively. **j** Exemplary still images from time courses between NEBD and anaphase of eGFP-cyclin A2 destruction in HEK cells of wild-type cyclin A2 and the ΔD2 mutant. The chromosomes are coloured in magenta and cyclin A2 in green, with the outline of the cells marked with dashed yellow lines. Time is given as hh:mm. Scale bar 10 µm. Degradation of cyclin A2 is slightly delayed in the absence of the D2 box (compare the two cells at 15 and 21 min). The complete time courses are shown as Supplementary Movies 1 and 2. Source data are provided as a Source Data file

disrupted (ΔKD1A). Strikingly, this ΔKD1A mutant was efficiently ubiquitinated by APC/C$^{Cdc20}$ to a similar extent as wild-type cyclin A2 (Fig. 2a, compare lanes 1 and 4). However, unlike wild-type cyclin A2, ubiquitination of the cyclin A2$^{ΔKD1A}$ mutant was virtually abolished in the presence of the MCC, indicating a loss of resistance to the MCC (Fig. 2a, compare lanes 2, 3 and 5, 6). In vivo, degradation of the ΔKD1A mutant was delayed to anaphase during an unperturbed mitosis, reminiscent of wild-type cyclin B1 (Fig. 2c, Supplementary Fig. 3c and Supplementary Movies 3, 4). Under a SAC arrest, the ΔKD1A mutant was partially stabilized until it was rapidly degraded by APC/C$^{Cdc20}$ when the SAC was inactivated using Mps1 kinase inhibitor reversine[42] (Fig. 2d). These findings suggested the possibility that an unidentified degron(s) is present in cyclin A2 to target its ubiquitination by APC/C$^{Cdc20}$, and contribute to cyclin A2 degradation in vivo.

**Identification of a non-canonical degron in cyclin A2.** Previous findings showed that deletion of the N-terminal 60 residues failed to stabilize cyclin A2, whereas deletion of the first 80 residues prevented its degradation[17]. This raised the possibility that the region between residues 60–80 may play a role in targeting cyclin A2 for degradation, although a canonical degron was not identified. To investigate whether a previously unidentified degron is present within this region, residues within 60–80 of cyclin A2 were replaced with GSA-linkers and combined with the ΔKD1A mutation. Strikingly, mutating residues 65–75 or 70–80 in addition to ΔKD1A completely abolished cyclin A2 ubiquitination by APC/C$^{Cdc20}$ (Fig. 2b, lanes 4 and 5). Furthermore, a peptide modelled on residues 60–80 suppressed ubiquitination of the cyclin A2 ΔKD1A mutant by APC/C$^{Cdc20}$ (Fig. 2e). These results indicated that an unidentified degron, lying within residues 70–80, contributes to the targeting of cyclin A2 for ubiquitination by APC/C$^{Cdc20}$. Combining the ΔKD1A mutation with mutation of residues 70–80 (designated D2, see below) (equal to Δall) also abolished the resistance of cyclin A2 ubiquitination to the MCC-imposed inhibition of APC/C activity (Fig. 1f, g lane 6).

In agreement with our in vitro analysis, mutation of residues 70–80 combined with the ΔKD1A mutation (Δall) completely stabilized cyclin A2 in vivo both in unperturbed and SAC arrested cells, leading to prolonged mitosis and disruption of cell division (Fig. 2c, d and Supplementary Fig. 3b–e). The unperturbed HEK cells expressing the cyclin A2 Δall mutant initially aligned their chromosomes, but failed to maintain a metaphase plate (Supplementary Fig. 3c and Supplementary Movie 5), consistent with previous results, suggesting a role for cyclin A2 in regulating kinetochore-microtubule attachments[15].

Residues 70–81 of cyclin A2 contains the sequence $^{72}VxxLxDLxx^{81}N$, similar to the consensus sequence of a canonical D box except for a Val substitution of Arg at position 1 (P1) (Fig. 2g). Nonetheless, exceptions for an Arg at the P1 residue have been observed for the D boxes of other APC/C substrates[9,41]. We named this new putative D box as the D2 box. The D2 box sequence is highly conserved among mammals ranging from human to mouse (Fig. 2f, g). Mouse and rat cyclin A2 have lost the D1 box and retained only the D2 box (Fig. 2f, g).

To verify whether the newly identified degron is indeed a D box, a competition assay was performed using a D box peptide modelled on the high affinity yeast substrate Hsl1[43,44]. Addition of the Hsl1 D box peptide potently inhibited ubiquitination of the cyclin A2 ΔKD1A mutant (with only the D2 box present) (Fig. 3a, lanes 4–6), indicating that both the Hsl1 D box peptide and the D2 box of cyclin A2 compete for the same binding site on APC/C$^{Cdc20}$. Moreover, point mutations of the consensus residues of the D2 box to alanines (D2$^{mut2}$) combined with the ΔKD1A mutation ablated cyclin A2 ubiquitination by APC/C$^{Cdc20}$ (Fig. 3b, c lane 3), reminiscent of mutating all four degrons (Δall) in cyclin A2 (Fig. 3b, c lane 4). The assignment of the identified degron as a D box is consistent with the finding that mutations in residues 81–90 containing an Asn at P10 reduced cyclin A2 ubiquitination (Fig. 2b, lane 6), as C-terminal hydrophilic residues of the D box are involved in interactions with Apc10[45–48]. In addition, the D2 box of cyclin A2 can function as a portable destruction signal in other APC/C substrates, as insertion of the D2 box into a securin mutant lacking its native D box rescued its ubiquitination by APC/C$^{Cdc20}$ (Fig. 3d).

**Cyclin A2 engages APC/C$^{Cdc20}$ in two different binding modes.** To identify whether the D2 box of cyclin A2 associates with APC/C$^{Cdc20}$ in the same way as a canonical D box, we determined the cryo-EM structure of APC/C$^{ΔApc1-300s}$ in complex with Cdc20 and wild-type Cdk2-cyclinA2-Cks2 to 3.2 Å resolution (Fig. 4a, Supplementary Figs. 4a–c, f, 5 and Supplementary Table 1). APC/C$^{ΔApc1-300s}$ is a constitutively active APC/C mutant that mimics wild-type phosphorylated APC/C[48]. Focused three-dimensional (3D) classification identified a large movement of the Cdc20 WD40 domain that is responsible for substrate recognition (Supplementary Fig. 5h), limiting its local resolution to 4.5–5.5 Å (Supplementary Fig. 5e, g). Notably, EM densities corresponding to the globular domains of Cdk2-cyclinA2-Cks2 were not visible, indicating their conformational flexibility. Two major 3D classes from the same dataset using wild-type Cdk2-cyclinA2-Cks2 as a substrate were characterized, corresponding to either the D1 box

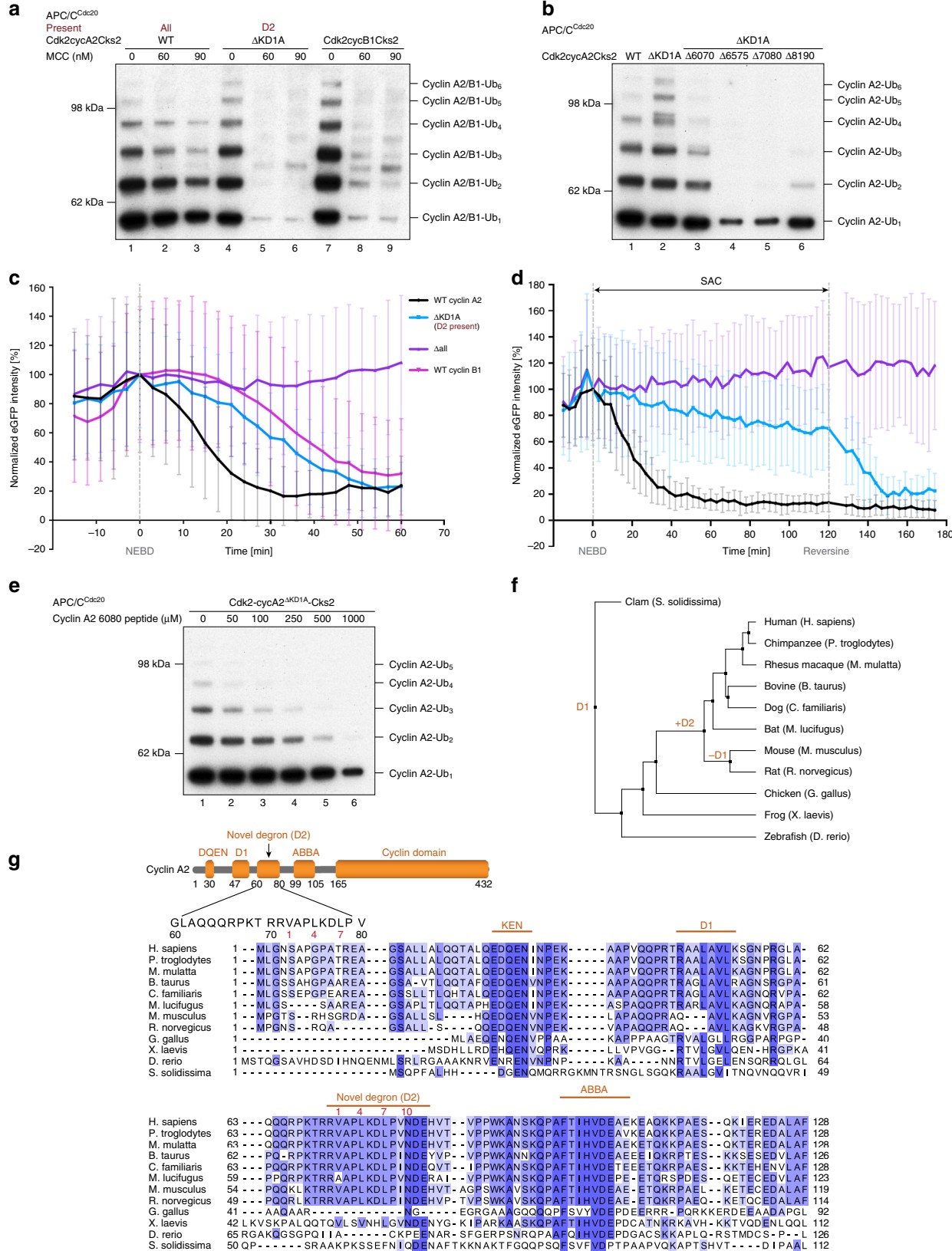

or D2 box bound to Cdc20[WD40] (Fig. 4b, d and Supplementary Fig. 5d, f). Assignment of the two 3D classes to the D1 and D2 boxes was based on a cryo-EM structure of APC/C[ΔΔApc1-300s] in complex with Cdc20 and Cdk2-cyclinA2[ΔD1]-Cks2 lacking the D1 box, determined to 3.7 Å resolution (Fig. 4c and Supplementary Fig. 4f). In this structure, only a C-shaped density connecting

Cdc20[WD40] and Apc10 was observed (Fig. 4c), identifying this density as the D2 box. In complexes of APC/C[Cdc20] with both wild-type Cdk2-cyclinA2-Cks2 and Cdk2-cyclinA2[ΔD1]-Cks2, the D2 box forms a C-shaped loop connecting Cdc20[WD40] and Apc10 (Fig. 4b, c), resembling the Hsl1 D box that is bound to APC/C[Cdc20] (ref. [48]) (Fig. 4g). In contrast, the D1 box class of

**Fig. 2** An unidentified degron conserved among mammals is present in cyclin A2. **a** Mutation of all three known degrons (KEN box, D1 box and ABBA motif) in cyclin A2 did not impair cyclin A2 ubiquitination by APC/C[Cdc20] (compare lanes 1 and 4). However, the mutant cyclin A2[ΔKD1A] is subject to MCC-imposed inhibition (lanes 4–6), similar to Cdk2-cyclinB1-Cks2 (lanes 7–9). The remaining degrons in cyclin A2 after the mutation are labelled in dark red. **b** Replacing overlapping segments within residues 60–80 of cyclin A2 with GSA-linkers identified that residues 70–80 may contain an unidentified degron. Mutation of either residues 65–75 or 70–80 in addition to ΔKD1A severely impaired cyclin A2 ubiquitination by APC/C[Cdc20]. **c, d** Degradation profiles of the eGFP-cyclin A2 mutants where either the KD1A degrons (leaving the D2 box present) or all four degrons were mutated in HEK cells under unperturbed mitosis (**c**) and a SAC arrest using STLC (**d**). Degradation pattern of cyclin B1 was recorded as a reference. The total eGFP fluorescence minus background fluorescence for each cell at each time point was normalized to NEBD, quantified and plotted over time. The timepoints of NEBD and reversine addition to release the SAC arrest are marked, respectively. Error bars indicate mean ± standard deviation of $n = 62$ wild-type cyclin A2, 24 ΔKD1A, 37 Δall and 45 wild-type cyclin B1 cells under unperturbed mitosis and 13 wild-type cyclin A2, 11 ΔKD1A and 19 Δall cells under a SAC arrest from three experiments. **e** Addition of a peptide modelled on residues 60–80 of cyclin A2 inhibited ubiquitination of the cyclin A2 ΔKD1A mutant, confirming that this region targets cyclin A2 for ubiquitination. **f, g** Sequence alignment of the N-terminal domain of cyclin A2 and phylogenic tree. Positions of the four degrons are indicated. All cyclin A2 sequences used in the sequence alignment contain the D1 box (except for mouse and rat), whereas the D2 box appeared at a later stage of evolution. Source data are provided as a Source Data file

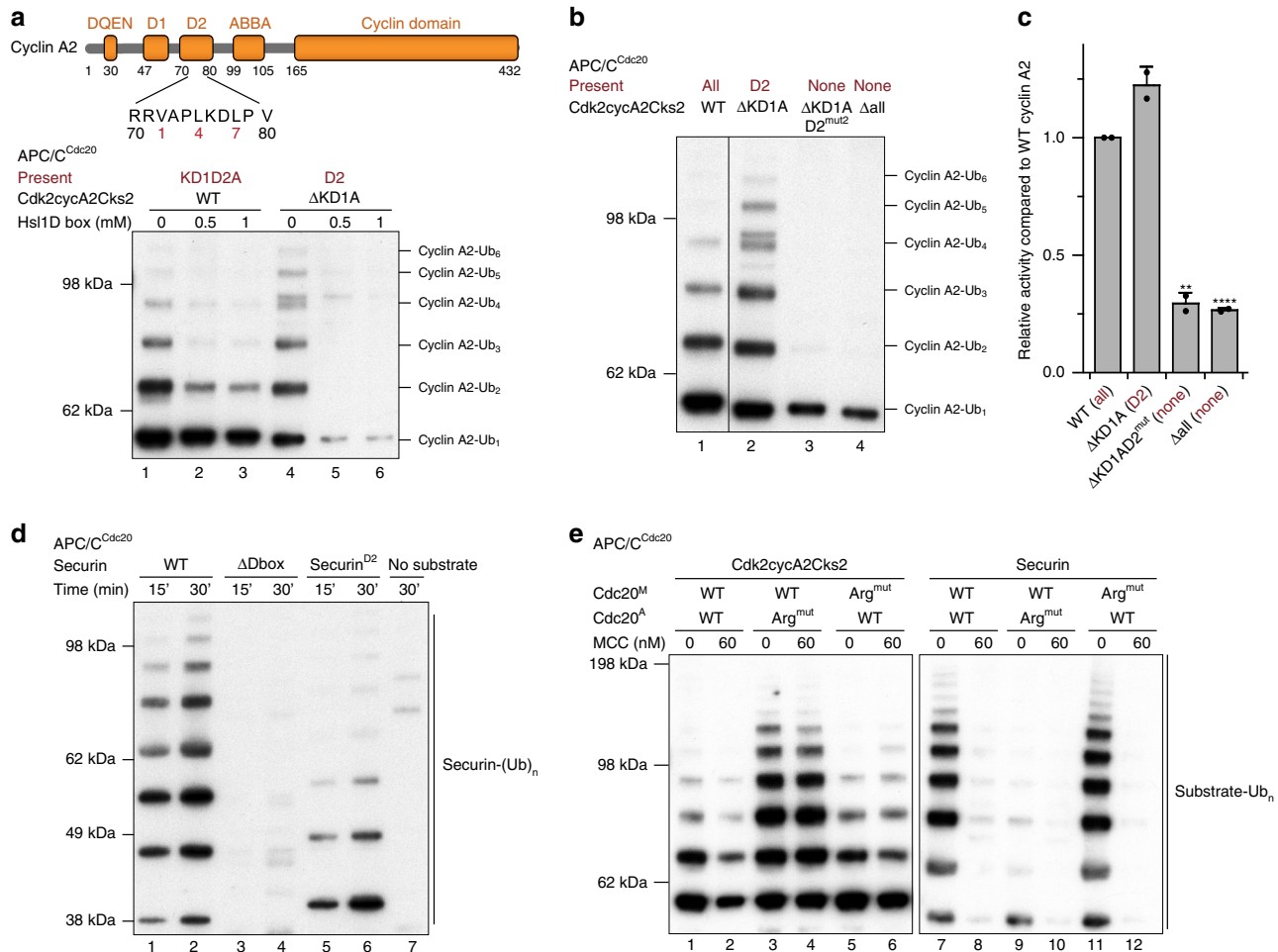

**Fig. 3** The identified degron in cyclin A2 is a non-canonical D box. **a** Addition of the Hsl1 D box peptide reduced ubiquitination of wild-type cyclin A2 (lanes 1–3) and potently suppressed cyclin A2 ubiquitination when only the D2 box was present (ΔKD1A, lanes 4–6). **b, c** The D2 box alone (ΔKD1A, lane 2) is sufficient to allow cyclin A2 ubiquitination by APC/C[Cdc20], yet additional single mutations at positions 1, 4 and 7 of the D2 box severely impaired its ubiquitination (ΔKD1A D2[mut2], lane 3), mimicking the effect when all four degrons were mutated (Δall, lane 4). The Western-blot was quantified to show the effect of individual mutations, with error bars indicating standard deviation. The APC/C activity towards cyclin A2 mutants is normalized to ubiquitination of wild-type cyclin A2 and significance is calculated using unpaired Student's $t$-test (indicated with stars, $n = 2$, Supplementary Table 2). **d** Insertion of the cyclin A2 D2 box into securin rescued its ubiquitination (securin[D2], lanes 5 and 6) from loss of its native D box (lanes 3 and 4). **e** Disrupting the P1 Arg-binding pocket for the D box (D177S&E465S) enhanced cyclin A2 ubiquitination by APC/C[Cdc20], but ablated securin ubiquitination, indicating that the D2 box binding leads to more efficient ubiquitination than the D1 box. Source data are provided as a Source Data file

wild-type Cdk2-cyclinA2-Cks2 in complex with APC/C[Cdc20] differs from the D2 box class due to a 20 Å shift of Cdc20[WD40], with a distinct long-kinked loop structure assigned to the D1 box (Fig. 4d, e and Supplementary Fig. 5h). Superposition of the two classes onto Cdc20[WD40] revealed that the N-terminal segment of both the D1 and D2 boxes are similar (Fig. 4e, f). However, unlike the D2 box, the C-terminal region of the D1 box folds back and is sandwiched between Cdc20[WD40] and Apc10.

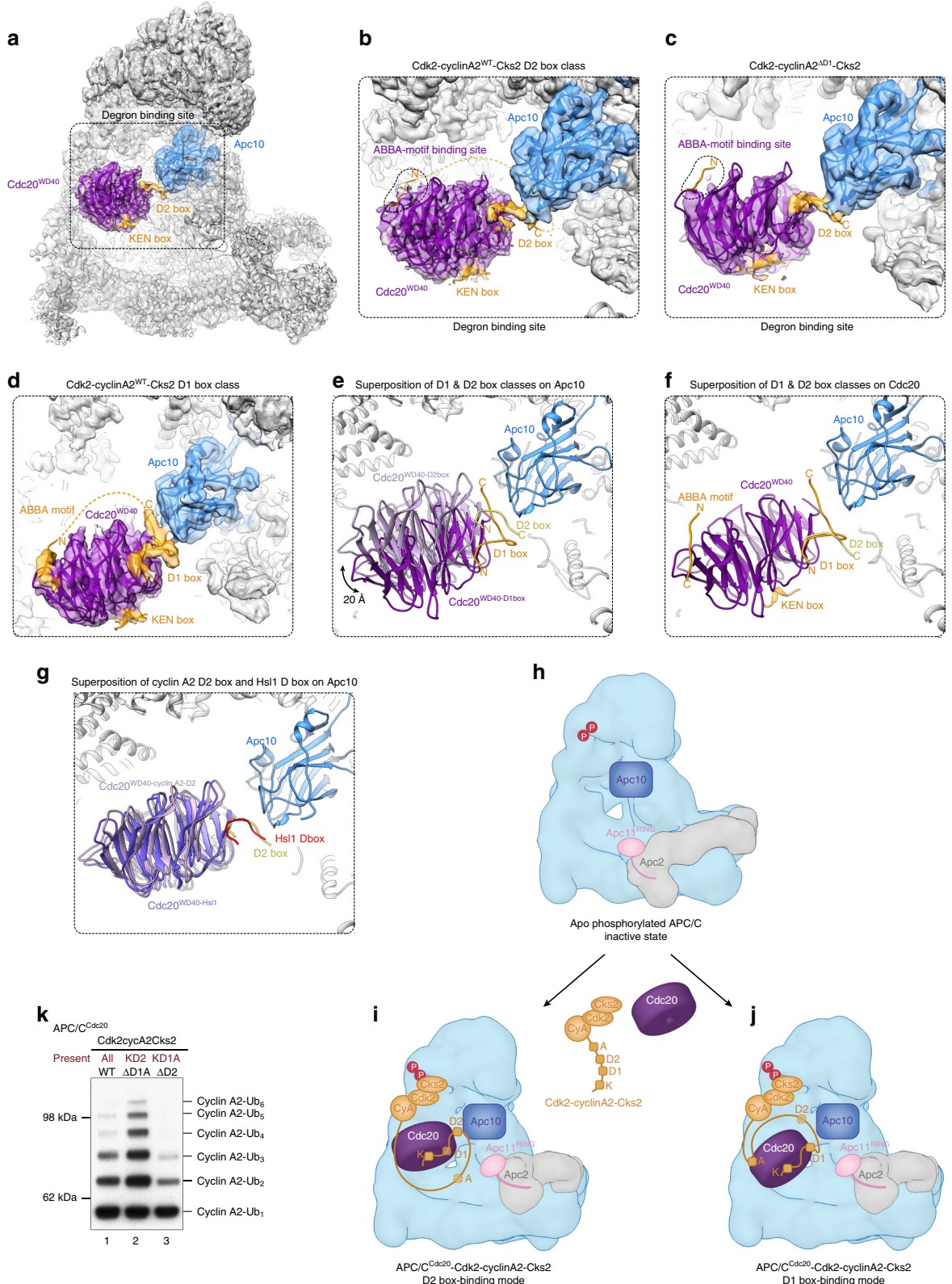

When the D2 box is engaged, weak densities accounting for the KEN box were observed on the top surface of Cdc20$^{WD40}$, whereas the ABBA-motif-binding pocket was unoccupied (Fig. 4b, c). A minimal distance of 60 Å required to bridge the D2 box and ABBA motif makes the ABBA-motif-binding pocket out of reach when the D2 box is bound (Fig. 4b). In addition, the C-terminus of the D2 box is directed away from the ABBA-motif-binding pocket due to Apc10 (Fig. 4b, c). Interestingly, strong EM densities for both the KEN box and ABBA motif were observed when the D1 box was bound (Fig. 4d). With a distance of 41 residues to the ABBA motif, the D1 box is within reach of the ABBA-motif-binding site. Furthermore, unlike the D2 box, the C-terminus of the D1 box is directed around Cdc20$^{WD40}$ towards the ABBA-motif-binding pocket (Fig. 4d, f). These observations

**Fig. 4** Cyclin A2 engages APC/C$^{Cdc20}$ with two binding modes. **a** Cryo-EM reconstruction of the constitutive active mutant APC/C$^{\Delta Apc1\text{-}300s}$ in complex with Cdc20 and Cdk2-cyclinA2-Cks2 at 3.2 Å resolution, with the D2 box and KEN box density highlighted in orange, Cdc20 in purple and Apc10 in blue. Individual chains of the APC/C$^{Cdc20\text{-}Hsl1}$ structure were fitted into the EM density (PDB 5G04[48]). Coordinates for Cdc20, Apc10 and the D2 box and KEN box of cyclin A2 are coloured according to the density, with the remaining APC/C subunits in grey. **b** Close-up view of the degron-binding site when the D2 box occupies the D-box binding site. No densities corresponding to the ABBA motif could be found in its binding pocket (dashed black line). **c** Close-up view of the degron-binding site when the ΔD1 mutant was used as a substrate which closely resembles the D2 box class of the wild-type structure. **d** Close-up view of the degron-binding site in the D1 box class when wild-type Cdk2-cyclinA2-Cks2 was used as a substrate reveals that the D1 box binds to APC/C$^{Cdc20}$ in cooperation with the KEN box and ABBA motif. **e** Superposition of the D1 box and D2 box classes on Apc10 revealed a large movement of Cdc20$^{WD40}$. The D1 box is depicted in orange while the D2 box in yellow. Cdc20$^{WD40}$ from the D1 box class is in dark purple, while that of the D2 box class in pale purple. **f** Superposition of the two classes on Cdc20$^{WD40}$ showed that the D2 box overlaps well with the N-terminal region of the D1 box, yet the C-terminal segment of the D1 box folds back with a kink to be sandwiched between Cdc20$^{WD40}$ and Apc10. **g** Superposition of the cyclin A2 D2 box structure with the structure of the canonical D box of Hsl1 (PDB 5G04[48]). **h–j** Schematic cartoon showing the two different binding modes of cyclin A2 to APC/C$^{Cdc20}$. **k** Ubiquitination assay showing different ubiquitination efficiency of the two binding modes of cyclin A2 to APC/C$^{Cdc20}$. Source data are provided as a Source Data file

indicate that cyclin A2 can engage APC/C$^{Cdc20}$ with two distinct binding modes (Fig. 4h–j).

**The D2 box of cyclin A2 is critical for its ubiquitination.** The newly identified D2 box is important for cyclin A2 ubiquitination by APC/C$^{Cdh1}$, APC/C$^{Cdc20}$ and APC/C$^{MCC}$, as mutating the D2 box alone (ΔD2) resulted in a reduced level of cyclin A2 ubiquitination in all three instances (Fig. 1b–g). Specifically, the cyclin A2 ΔD2 mutant was the only single degron mutation that resulted in a reduction of cyclin A2 ubiquitination by APC/C$^{Cdc20}$ (Fig. 1d, e and Table 1). In agreement with the in vitro assays, live cell imaging showed that relative to other single degron mutations, disruption of the D2 box caused the longest delay in cyclin A2 degradation (to around metaphase) (Fig. 1h–j). Nevertheless, the single absence of the D2 box did not fully stabilize cyclin A2 in cells (Fig. 1h, i). This indicates that in addition to the D2 box, the other three cyclin A2 degrons contribute to its degradation in vivo.

To explore the roles of individual degrons, we performed a comprehensive analysis of degron mutations in cyclin A2 (Table 1). We found that the newly identified D2 box contributes to the major activity of APC/C$^{Cdc20}$, augmented by the KEN box (Fig. 5a, b and Supplementary Fig. 2b). A cyclin A2 mutant with only the D2 box present (ΔKD1A) is ubiquitinated by APC/C$^{Cdc20}$ to the same extent as wild-type cyclin A2, while addition of the KEN box slightly enhanced its ubiquitination (Fig. 5a, b, compare lanes 1, 4 and 7). In contrast to APC/C$^{Cdc20}$, the KEN box plays a major role for cyclin A2 ubiquitination by APC/C$^{Cdh1}$, augmented by the D2 box (Supplementary Fig. 1g, h, compare lanes 1 and 7, 2a).

Interestingly, combining the D1 box with either the ABBA motif or the KEN box allowed cyclin A2 ubiquitination at a low level (Fig. 5a, b lanes 6 and 10), indicating that in the absence of D2, D1 cooperates with the ABBA motif and KEN box to promote cyclin A2 ubiquitination by APC/C$^{Cdc20}$. This result is consistent with the cryo-EM reconstructions of the APC/C$^{\Delta Apc1\text{-}300s\text{-}Cdc20}$-Cdk2-cyclinA2-Cks2 complex revealing a mixed population of conformations, with either D1 or D2 interacting with APC/C$^{Cdc20}$ (Fig. 4b, d). In support of this idea, disrupting the P1-Arg-binding site of the Cdc20 D-box receptor (Arg$^{mut}$) resulted in enhanced cyclin A2 ubiquitination by APC/C$^{Cdc20}$, while it strongly impaired securin ubiquitination (Fig. 3e, compare lanes 1–3 and 7–9). Disruption of the P1-Arg-binding site would favour the engagement of D2, leading to enhanced ubiquitination of cyclin A2. Therefore, two mutually exclusive binding modes allow cyclin A2 ubiquitination by APC/C$^{Cdc20}$ (Fig. 4h–j), resulting in different efficiency of cyclin A2 ubiquitination (Fig. 4k). The D2 box is self-sufficient for cyclin A2 ubiquitination and is augmented by the KEN box (Fig. 4i),

**Table 1 Summary of the effects of cyclin A2 degron mutations to its ubiquitination**

| Degron mutation | APC/C$^{Cdc20}$ | APC/C$^{MCC}$ | Degrons present |
|---|---|---|---|
| *KEN box mutations* | | | |
| ΔK alone | Same as WT | Lower | D1D2A |
| ΔKD1 | Same as WT | Lower | D2A |
| ΔKD2 | Lower | Lower | D1A |
| ΔKA | Same as WT | No activity | D1D2 |
| ΔKD1D2 | No activity | No activity | A |
| ΔKD1A | Same as WT | No activity | D2 |
| ΔKD2A | No activity | No activity | D1 |
| ΔKD1D2A | No activity | No activity | None |
| *D1 box mutations* | | | |
| ΔD1 alone | Higher | Same as WT | KD2A |
| ΔKD1 | Same as WT | Lower | D2A |
| ΔD1D2 | No activity | No activity | KA |
| ΔD1A | Higher | Lower | KD2 |
| ΔKD1D2 | No activity | No activity | A |
| ΔKD1A | Same as WT | No activity | D2 |
| ΔKD2A | No activity | No activity | K |
| ΔKD1D2A | No activity | No activity | None |
| *D2 box mutations* | | | |
| ΔD2 alone | Lower | Lower | KD1A |
| ΔKD2 | Lower | Lower | D1A |
| ΔD1D2 | No activity | No activity | KA |
| ΔD2A | Lower | No activity | KD1 |
| ΔKD1D2 | No activity | No activity | A |
| ΔKD2A | No activity | No activity | D1 |
| ΔD1D2A | No activity | No activity | K |
| ΔKD1D2A | No activity | No activity | None |
| *ABBA motif mutations* | | | |
| ΔA alone | Higher | Lower | KD1D2 |
| ΔKA | Same as WT | No activity | D1D2 |
| ΔD1A | Higher | Lower | KD2 |
| ΔD2A | Lower | No activity | KD1 |
| ΔKD1A | Same as WT | No activity | D2 |
| ΔKD2A | No activity | No activity | D1 |
| ΔD1D2A | No activity | No activity | K |
| ΔKD1D2A | No activity | No activity | None |

Any combinations with the D2 box mutated resulted in a reduction or elimination of cyclin A2 ubiquitination by both APC/C$^{Cdc20}$ and APC/C$^{MCC}$

whereas the weak D1 box requires cooperative binding of the KEN box and ABBA motif (Fig. 4j).

In the presence of the MCC, none of the four identified degrons alone (KEN, D1, D2, ABBA) was sufficient to promote cyclin A2 ubiquitination in vitro (Fig. 5c, d, compare lanes 1–5

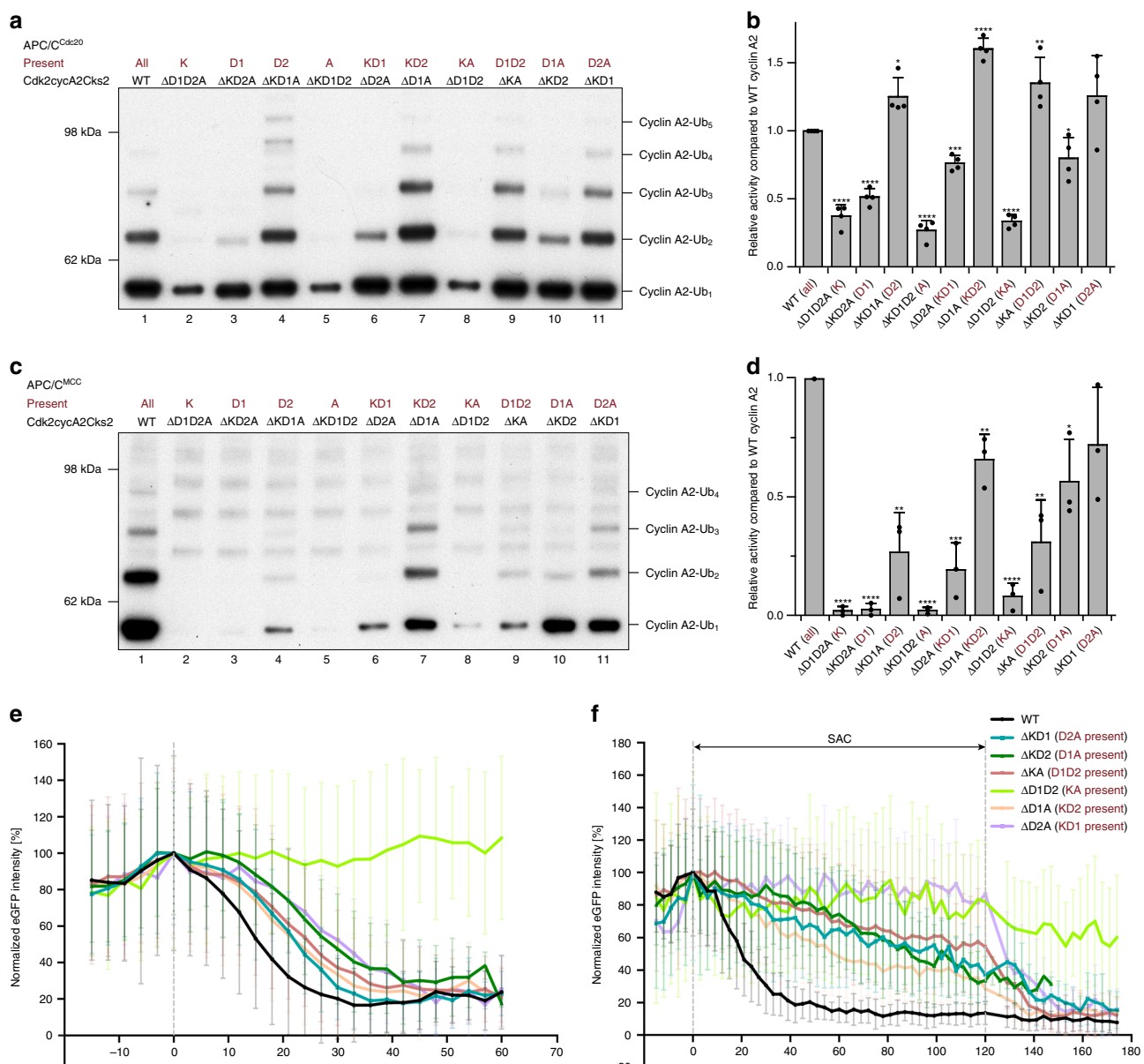

**Fig. 5** A cooperation of multiple degrons confers the resistance of cyclin A2 to MCC-imposed inhibition. **a**–**d** Ubiquitination assays of double and triple degron mutations in cyclin A2 by APC/C^Cdc20 (**a**) and APC/C^MCC (**c**). In all assays, cyclin A2 is in complex with Cdk2-Cks2. The D2 box is the critical determinant for cyclin A2 ubiquitination by APC/C^Cdc20 (compare lanes 1, 4, 7, 9 and 11 in **a**). To resist MCC-mediated inhibition, cyclin A2 requires both the KEN box and ABBA motif together with the D2 box (compare lanes 1, 7 and 11 in **c**). The remaining degrons in cyclin A2 after the mutation are labelled in dark red. Ubiquitination reactions were analysed by western blotting with an anti-His antibody to detect the His-tag of the ubiquitin-modified substrates. Control gels indicating the unmodified substrate for representative reactions are shown in Supplementary Fig. 2. The western-blots were quantified to show the effect of individual mutations, with error bars indicating standard deviation. The APC/C activity towards cyclin A2 mutants is normalized to ubiquitination of wild-type cyclin A2 and significance is calculated using unpaired Student's t-test (indicated with stars, $n = 4$ for **b** and 3 for **d**, Supplementary Table 2). **e**, **f** Degradation profiles of the eGFP-cyclin A2 double degron mutants in HEK cells under unperturbed mitosis (**e**) and a SAC arrest using STLC (**f**). The total eGFP fluorescence minus background fluorescence for each cell at each time point was normalized to NEBD, quantified and plotted over time. The timepoints of NEBD and reversine addition to release the SAC arrest are marked, respectively. Error bars indicate mean ± standard deviation of $n = 62$ wild-type, 33 ΔKD1, 58 ΔKD2, 34 ΔKA, 28 ΔD1D2, 48 ΔD1A and 23 ΔD2A cells under unperturbed mitosis and 13 wild-type, 15 ΔKD1, 12 ΔKD2, 20 ΔKA, 12 ΔD1D2, 19 ΔD1A and 13 ΔD2A cells under a SAC arrest from three experiments. Source data are provided as a Source Data file

and Supplementary Fig. 2c). The D2 box is the dominating determinant, and addition of either the KEN box or the ABBA motif to D2 resulted in cyclin A2 ubiquitination by APC/C^MCC that approached wild-type levels (Fig. 5c, d, compare lanes 1, 7 and 11). Consistent with the in vitro assays, a combination of D2 with either the KEN box or ABBA motif advanced cyclin A2 degradation in HEK cells to an earlier time point compared to

when D2 was present alone, during both an unperturbed mitosis and a SAC arrest (Fig. 2c, d and Fig. 5e, f). These results suggest that the KEN box and ABBA motif work cooperatively with D2 to allow cyclin A2 to overcome the MCC-imposed inhibition of APC/C activity.

A dominant role for the D2 box raised the question of why human cyclin A2 retained the D1 box. A titration of the cyclin A2

ΔD1 mutant revealed that at lower concentrations (0.18–0.48 μM), disrupting D1 reduced cyclin A2 ubiquitination by APC/C$^{MCC}$ (Supplementary Fig. 1i, j compare lanes 1 and 2 with 5 and 6), consistent with the observation that D1 contributes to cyclin A2 degradation during a SAC arrest in HEK cells (Fig. 1i and Supplementary Fig. 3e). These results imply that D1 may increase the avidity of cyclin A2 as a substrate to APC/C$^{MCC}$.

**Cyclin A2 is targeted by the repressed APC/C$^{MCC}$.** It had previously been suggested that the ABBA motif of cyclin A2 confers resistance to the MCC-mediated inhibition of APC/C$^{Cdc20}$ by outcompeting BubR1, thereby allowing its binding to the activating Cdc20$^A$ (Cdc20 of the APC/C) subunit to be ubiquitinated by APC/C$^{Cdc20}$ (ref. [6,24]). To test this hypothesis, we incubated pre-formed and purified APC/C$^{MCC}$ with wild-type Cdk2-cyclinA2-Cks2 and analysed the complex by size exclusion chromatography. A complex of APC/C$^{MCC}$-Cdk2-cyclinA2-Cks2 was formed, with no evident dissociation of the MCC from APC/C$^{Cdc20}$ (Fig. 6a–d). This result suggests an alternative model in which the MCC and Cdk2-cyclinA2-Cks2 simultaneously interact with APC/C$^{Cdc20}$. In this model, cyclin A2 binds to and is ubiquitinated by the repressed APC/C$^{MCC}$. Mutation of the cyclin A2 ABBA motif resulted in slightly reduced binding of Cdk2-cyclinA2$^{ΔABBA}$-Cks2 to APC/C$^{MCC}$ (Supplementary Fig. 6a), consistent with our findings that the ABBA motif is required to confer cyclin A2's resistance to the MCC-imposed inhibition of APC/C activity.

Within APC/C$^{MCC}$, there are two Cdc20 subunits[21–23] and therefore, two binding sites for each degron (Fig. 7a). To determine which of the two Cdc20 subunits engages cyclin A2's ABBA motif, mutations (I235S and Y279E)[6] were introduced into Cdc20 to disrupt the ABBA-motif-binding pocket (Supplementary Fig. 6b) on either Cdc20$^A$ (Cdc20 of the APC/C) or Cdc20$^M$ (Cdc20 of the MCC). Strikingly, disrupting the ABBA-motif-binding site on Cdc20$^M$ strongly impaired the resistance of cyclin A2 to MCC-imposed inhibition of APC/C activity (Fig. 6e, f, lanes 7–9), whereas the equivalent mutation in Cdc20$^A$ did not affect cyclin A2 ubiquitination in the presence of the MCC (Fig. 6e, f, lanes 4–6). The opposite effect was observed for securin where mutation of Cdc20$^M$ weakened the inhibitory strength of the MCC to securin ubiquitination (Supplementary Fig. 6c). Therefore, the loss of resistance to MCC-imposed inhibition caused by disrupting the ABBA-motif-binding pocket in Cdc20$^M$ is unique to cyclin A2, indicating that the ABBA motif of cyclin A2 interacts with Cdc20$^M$. Similarly, a mutation disrupting the interactions of hydrophobic residues within the D-box binding pocket (L176A) revealed that the D2 box of cyclin A2 engages the Cdc20$^M$ subunit of APC/C$^{MCC}$ (Fig. 6g, h and Supplementary Fig. 6d).

An attempt to determine the cryo-EM structure of the APC/C$^{MCC}$-Cdk2-cyclinA2-Cks2 complex did not reveal additional densities accounting for Cdk2-cyclinA2-Cks2 (Supplementary Figs. 4d–f, 6e and Supplementary Table 1). Thus we cannot distinguish the degrons bound to the APC/C$^{MCC}$ degron recognition sites are derived from the MCC or cyclin A2. Nonetheless, a four-fold increase in the ratio between the open and closed states of APC/C$^{MCC}$ was observed in the presence of Cdk2-cyclinA2-Cks2 compared to a control dataset collected for APC/C$^{MCC}$ alone (Supplementary Fig. 6e). With this observation we speculate that Cdk2-cyclinA2-Cks2 may bind to APC/C$^{MCC}$ and induce an open conformation of APC/C$^{MCC}$ to enable cyclin A2 ubiquitination. When in the open state, the catalytic module of the APC/C (Apc2-Apc11) is accessible to engage the E2[22,23].

**Cooperative binding of multiple degrons allows cyclins to resist the SAC.** Our results reveal that the resistance of cyclin A2 to MCC-imposed inhibition of APC/C activity is conferred by cooperative binding of four factors: the KEN box, D2 box, ABBA motif and its associated Cks (Fig. 7a, b and Table 1). To confirm that the major determinant of SAC-resistant cyclin A2 degradation is the KEN-Dbox-ABBA cassette, we tested if we could convert the metaphase substrate cyclin B1, whose degradation is suppressed by the MCC, into a prometaphase substrate that can be ubiquitinated by APC/C$^{MCC}$. Cyclin B1 lacks both the KEN box and ABBA motif, as mutation of a putative KEN box (NAEN) in cyclin B1 failed to impair its ubiquitination by either APC/C$^{Cdh1}$ or APC/C$^{Cdc20}$, and cyclin B1 ubiquitination remained repressed by the MCC (Supplementary Fig. 6f). Insertion of the cyclin A2 KEN box alone failed to enhance cyclin B1 ubiquitination by APC/C$^{MCC}$ (Fig. 7c, d, compare lanes 2, 4 and 6). On the other hand, insertion of the ABBA motif into cyclin B1 conferred partial resistance of cyclin B1 to MCC-mediated inhibition (Fig. 7c, d, lanes 7 and 8). Insertion of both the KEN box and ABBA motif into cyclin B1 allowed for its ubiquitination by APC/C$^{MCC}$, with a relative activity comparable to that of wild-type cyclin A2 (Fig. 7c, d, compare lanes 9 and 10 to 11 and 12).

## Discussion

A long-standing puzzle has been how the cell-cycle-dependent change of substrate selection by the APC/C is achieved, in particular, how cyclin A2 evades the SAC to allow its degradation at prometaphase, whereas cyclin B1 and securin remain stable. Progression from prometaphase to metaphase requires destruction of cyclin A2, as persistent levels of cyclin A2 prevent stable kinetochore-microtubule attachments, thereby disrupting faithful chromosome segregation[15]. Despite its importance, the mechanistic details of how cyclin A2 resists the MCC-imposed inhibition of APC/C activity and the roles of individual degrons in cyclin A2 remained to be defined. In this study we identified a non-canonical D box (D2 box) in cyclin A2 that is critical for its efficient ubiquitination. Mutation of the D2 box reduced cyclin A2 ubiquitination by APC/C$^{Cdh1}$, APC/C$^{Cdc20}$ and APC/C$^{MCC}$ (Table 1). Moreover, it is the only degron that is self-sufficient to allow cyclin A2 ubiquitination by APC/C$^{Cdc20}$. To overcome MCC-imposed inhibition, cyclin A2 requires not only the previously characterized ABBA motif, but also its KEN box and D2 box.

In addition, we found that Cdk2-cyclinA2-Cks2 forms a stable complex with APC/C$^{MCC}$, without dissociating the MCC. Within APC/C$^{MCC}$, BubR1 blocks all six degron-binding sites on both Cdc20 subunits with two copies of the KEN box, D box and ABBA motif[22,23] (Fig. 7a, b). Our data suggest that the ABBA motif of cyclin A2 associates with Cdc20$^M$ by displacing the second ABBA motif (A2) of BubR1. With its strong ABBA motif sequences (Fig. 2g), cyclin A2 is capable of competing off the weaker A2 motif of BubR1[6]. This is consistent with previous results that mutation of the first ABBA motif (A1) in BubR1, that interacts with Cdc20$^A$, dissociated the MCC from the APC/C[49], making it unlikely that cyclin A2 would replace this ABBA motif.

The KEN-box binding site on Cdc20$^M$ is buried at the BubR1, Cdc20, Mad2 interface of the MCC[22,23,50] (Fig. 7a, b). Mutating the first KEN box (K1) of BubR1 that interacts with Cdc20$^M$ completely abolished formation of the MCC[49,51]. On the other hand, the KEN-box binding site on Cdc20$^A$ is more accessible and has the potential to accommodate the KEN box of cyclin A2. Cyclin A2 contains a strong KEN box with the sequence EDQENxxP (Fig. 2g) due to the Asp and Glu residues at P-1 and P-2 and C-terminal proline residue[9]. None of the two KEN boxes in BubR1 contains the preferred acidic residues at P-1 and the C-terminal proline, implying that cyclin A2 KEN box may have a higher affinity for the binding pocket, thereby displacing the KEN box (K2) of BubR1.

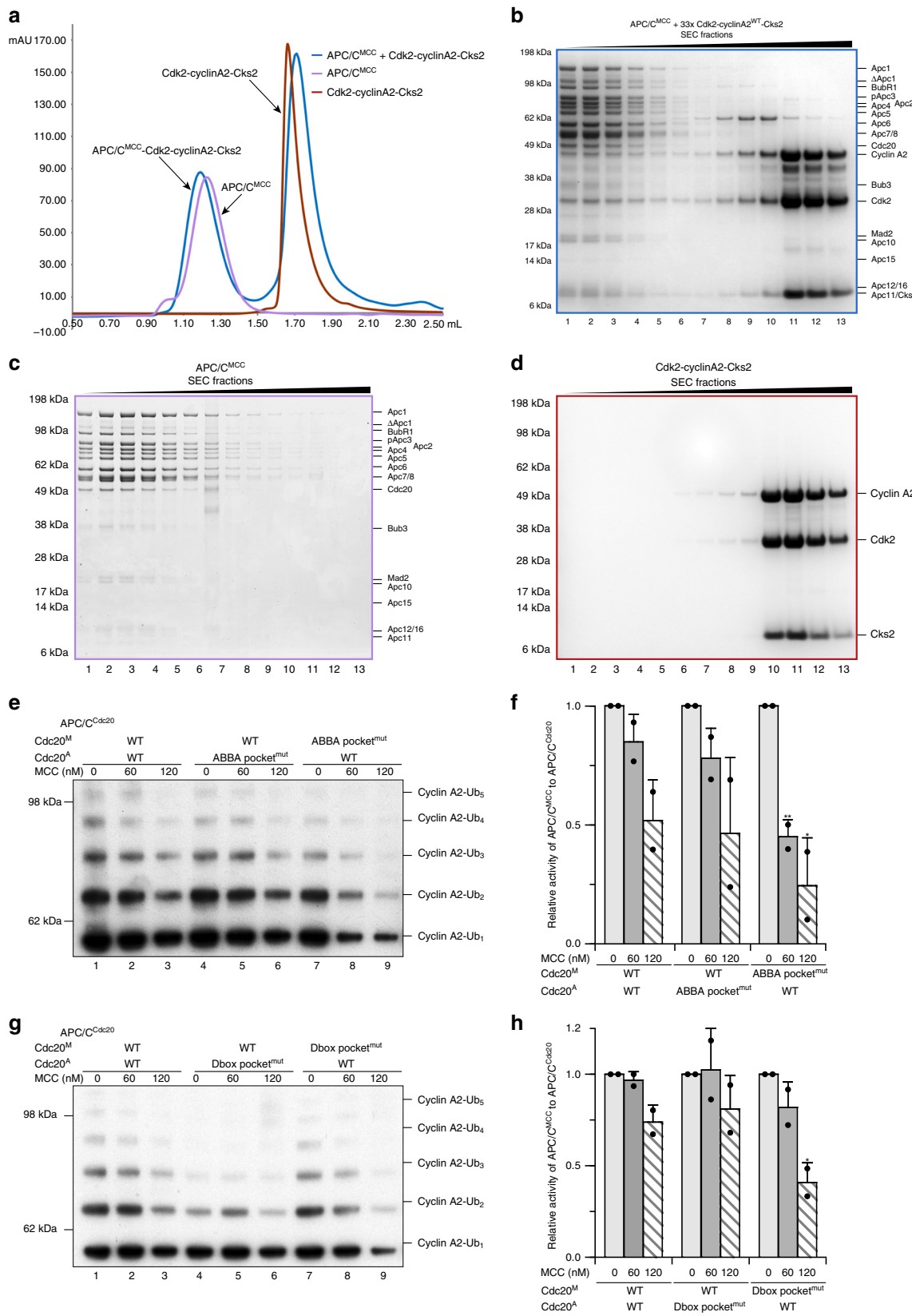

We showed that cyclin A2 interacts the D-box binding site of Cdc20$^M$ within APC/C$^{MCC}$. The D-box binding pocket on Cdc20$^M$ is exposed at the periphery, whereas that on Cdc20$^A$ is deeply buried between Cdc20$^A$ and the APC/C[22,23] (Fig. 7a, b). Association of the D2 box of cyclin A2 with Cdc20$^M$ would allow the N-terminus of cyclin A2 to bridge both Cdc20$^A$ and Cdc20$^M$ subunits, through its KEN box and ABBA motif, respectively (Fig. 7a, b). This cyclin A2-mediated bridging of APC/C$^{Cdc20}$ with the MCC may stabilize APC/C$^{MCC}$ and compensate for the disruption of BubR1 contacts with both Cdc20$^A$ and Cdc20$^M$.

**Fig. 6** Cdk2-cyclinA2-Cks2 associates with APC/C$^{MCC}$, with its ABBA motif and D2 box binding to Cdc20$^M$. **a–d** Size exclusion chromatography showed formation of a stable APC/C$^{MCC}$-Cdk2-cyclinA2-Cks2 complex with a peak shifted to higher molecular weight (blue line) compared to either APC/C$^{MCC}$ (purple line) or Cdk2-cyclinA2-Cks2 alone (red line). Cdk2-cyclinA2-Cks2 was added at a molar ratio of 33:1 to APC/C$^{MCC}$, where it could efficiently resist MCC-mediated inhibition in the ubiquitination assays. The fractions of APC/C$^{MCC}$-Cdk2-cyclinA2-Cks2 (**b**), APC/C$^{MCC}$ alone (**c**) and Cdk2-cyclinA2-Cks2 alone (**d**) were analysed on SDS-PAGE. **e, f** Disruption of the ABBA-motif-binding pocket on Cdc20 (with point mutations I235S&Y279E) of the MCC (Cdc20$^M$) strongly reduced the resistance of cyclin A2 to MCC-imposed inhibition of APC/C activity (lanes 7–9), whereas the same mutation on Cdc20$^A$ had no effect (lanes 4–6). **g, h** Disruption of the hydrophobic interactions within the D-box binding site (L176A) on Cdc20$^M$ resulted in increased inhibition of cyclin A2 ubiquitination by APC/C$^{MCC}$ (lanes 7–9). The histogram of relative APC/C$^{MCC}$ activity is normalized to respective APC/C$^{Cdc20}$ activity, with error bars indicating standard deviation, and significance is calculated using unpaired Student's $t$-test (indicated with stars, $n = 2$ for **f** and **h**, Supplementary Table 2). Source data are provided as a Source Data file

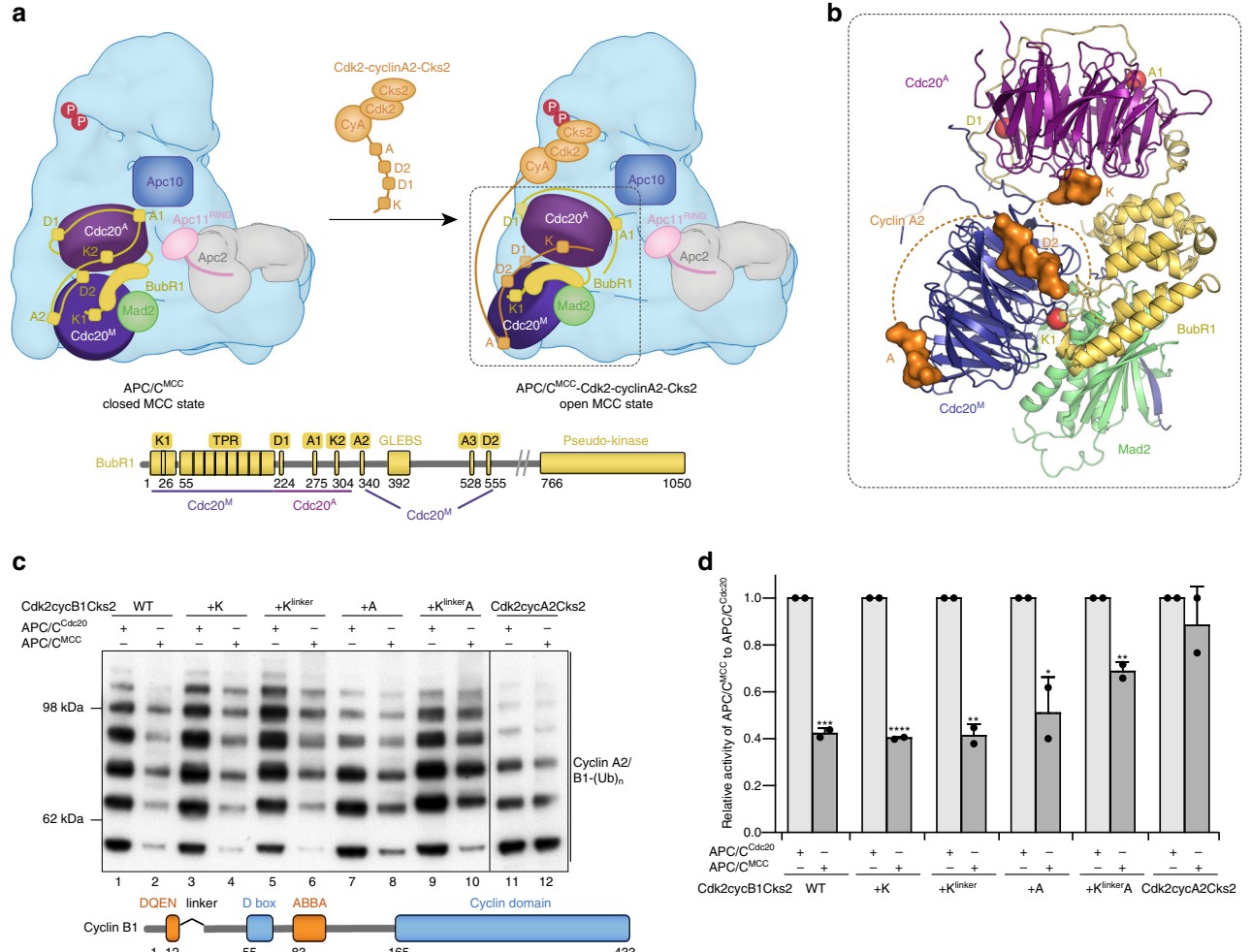

**Fig. 7** Mechanism of how cyclin A2 evades the SAC and conversion of cyclin B1 into a prometaphase substrate. **a** Schematic cartoon showing that Cdk-cyclinA2-Cks2 associates with APC/C$^{MCC}$ and displaces the ABBA motif (A2) and the D box (D2) of BubR1 from Cdc20$^M$. The more exposed KEN-box binding site on Cdc20$^A$ is likely to accommodate the KEN box of cyclin A2. Binding of cyclin A2 may induce the open state MCC. We propose that the D2 box is the principle D box; however, the D1 box augments affinity through an avidity effect. Bottom: Schematic of BubR1 domain organization showing its degrons and the respective interacting Cdc20 subunits. **b** Close-up view of a model with cyclin A2 associating with Cdc20$^A$-MCC (PDB 5LCW[22]). Cyclin A2 degrons are shown modelled as surface representations in orange occupying the more exposed degron-binding sites based on the equivalent pseudo-degrons of BubR1[22]. The remaining three degron-binding sites where BubR1 degrons could still engage are highlighted as red spheres (A1, D1, K1). **c, d** Insertion of both the KEN box and ABBA motif into cyclin B1 allowed it to resist MCC-imposed inhibition (compare lanes 1, 2 and 9, 10). Cyclin B1 is in complex with Cdk2-Cks2. The histogram of relative APC/C$^{MCC}$ activity is normalized to respective APC/C$^{Cdc20}$ activity, with error bars indicating standard deviation, and significance is calculated using unpaired Student's $t$-test (indicated with stars, $n = 2$, Supplementary Table 2). Source data are provided as a Source Data file

The additional affinity for APC/C$^{MCC}$ provided by the D1 box of cyclin A2 is likely mediated by avidity effects, providing insights into the role of multiple copies of the same degron present in some APC/C substrates. The metaphase substrates cyclin B1, which lacks both a functional KEN box and an ABBA motif, and securin, which lacks the ABBA motif and association of the Cks subunit, are unable to engage APC/C$^{MCC}$ and are thus stabilized during an active SAC.

Surprisingly, cyclin A2 associates with APC/C$^{Cdc20}$ through two distinct binding modes; a high activity mode mediated by its KEN and D2 boxes (Fig. 4i), and a lower activity mode via its KEN and D1 boxes and ABBA motif (Fig. 4j). Differences in the accessibility of lysines to the catalytic module between these two modes, or the distinct conformational states of Cdc20$^{WD40}$, may account for the different efficiency of cyclin A2 ubiquitination associated with these two binding modes. Our findings reveal that a single substrate can engage the APC/C through multiple binding modes to influence its timing and efficiency of ubiquitination. In principle, controlling which binding mode is adopted, possibly by restricting access to either the D1 or D2 box, would regulate the rate of cyclin A2 ubiquitination.

## Methods

**Expression and purification of human APC/C, MCC and coactivators.** The genes for recombinant human APC/C were cloned into a modified MultiBac system, expressed and purified as described[52]. The C-terminus of Apc4 was fused to a TEV (tobacco etch virus)-cleavable StrepIIx2 tag. In brief, recombinant APC/C expressed in High Five cells are purified with StrepTactin Superflow Cartridges (Qiagen) and eluted with 2.5 mM desthiobiotin (Santa Cruz). Eluted fractions from StrepTactin columns were incubated with TEV protease overnight at 4 °C and subsequently applied onto a 6 ml Resource Q anion-exchange column (GE Healthcare). Peak fractions after Resource Q were concentrated and further purified by a Superose 6 10/300 GL column (GE Healthcare).

Human Cdc20, Cdh1 and MCC were expressed and purified as described[22,46,48]. Full-length human Cdc20 was cloned into a modified FastBac HTa vector with an N-terminal His-MBP tag and expressed in Sf9 cells. Cdc20 was purified with a HisTrap HP column (GE Healthcare). Following TEV-cleavage overnight, the protein was re-applied onto the HisTrap HP column and the flow-through was collected. Full-length human Cdh1 contains an N-terminal His-tag and was expressed in High Five cells. Cdh1 was purified using the same procedure as Cdc20 except for an additional final purification step with the Superdex200 16/60 column (GE Healthcare). MCC was purified using a similar procedure as the APC/C with a StrepTactin Superflow cartridge (Qiagen) and a Superose 6 10/300 GL column (GE Healthcare).

Mutations in the degron-binding pockets of Cdc20 were introduced by site-directed mutagenesis. Two ABBA-motif-binding site mutants Cdc20$^{I235S\&Y279E}$ and Cdc20$^{I235S\&V295K}$ were generated. Two D-box binding pocket mutants were generated, with Cdc20$^{D177S\&E465S}$ disrupting the Arg-binding site and Cdc20$^{L176A}$ disrupting the hydrophobic binding pocket. The corresponding MCC mutants were produced by combining mutated Cdc20 with the wild-type BubR1, Bub3 and Mad2 proteins.

**Cloning, expression and purification of APC/C substrates.** Full-length human *cyclin A2* and *cyclin B1* were cloned into a pETM41 vector with an N-terminal His-MBP tag. The protein was expressed in either BL21 (DE3) Star cells for cyclin A2 or B834 (DE3) pLysS cells for cyclin B1 at 18 °C overnight. Full-length human *cyclin B1* additionally was cloned into a FastBac HTa vector using restriction enzyme cloning (NcoI and NotI) and expressed in Sf9 cells. Cdk2 phosphorylated at T160 was produced in *E. coli* by coexpression of human *GST–CDK2* and *S. cerevisiae GST–Cak1* (Cak1 is also called Civ1) as described[53] and D. Barford, J. Tucker, N.R. Brown and N. Hanlon (unpublished data). Pellets containing GST-3C-Cdk2, His-MBP-TEV-cyclin A2 or cyclin B1 and His-SUMO-TEV-Cks2 were co-lysed in the CDK lysis buffer (50 mM Tris/HCl pH 8.0, 200 mM NaCl, 5% glycerol and 2 mM DTT) supplemented with 0.1 mM PMSF, lysozyme, 5 units ml$^{-1}$ benzonase and Complete™ EDTA-free protease inhibitors (Roche). After sonication, the cells were centrifuged at 48,000 × g for 1 h at 4 °C and the supernatant was incubated with the Glutathione Sepharose™ 4B (GE Healthcare) for 3 h at 4 °C. The resin was washed with the CDK lysis buffer and the GST-tag of Cdk2 was cleaved off with 3C PreScission protease overnight at 4 °C. The flow-through from the resin was collected and TEV-cleaved overnight at 4 °C. Finally, the protein complex was purified by a Superdex 200 16/60 column (GE Healthcare) in the gel filtration buffer (20 mM Hepes pH 8.0, 150 mM NaCl, 0.5 mM TCEP).

The cyclin A2$^{\Delta K}$ mutant has the sequence DQEN replaced with Ala, while for the cyclin A2$^{\Delta D1}$ mutant the consensus residues at positions 1, 4 and 7 were mutated to Ala. For the cyclin A2$^{\Delta ABBA}$ mutant, the entire ABBA motif (PAFTIHVDE) was mutated to Ala. The cyclin A2$^{D2mut2}$ has the D2 box (VAPLKDL) mutated to AAPAKDA. The cyclin B1$^{\Delta'K'}$ mutant has the sequence NAEN replaced with Ala, whereas it was mutated to DQEN for the cyclin A2 KEN box insertion. A GSA-linker was inserted to match the amino acid distance between the KEN box and the D2 box in cyclin A2. The ABBA motif was inserted in cyclin B1 at the same amino acid distance as the D2 box and the ABBA motif in cyclin A2.

Full-length human securin was tagged with an N-terminal GST-tag and was expressed in BL21 (DE3) Star cells at 18 °C overnight. The cell pellets were lysed in CDK lysis buffer, following sonication the cleared lysate was incubated with the Glutathione Sepharose™ 4B (GE Healthcare) for 3 h at 4 °C. The resin was washed with the CDK lysis buffer and the GST-tag of securin was cleaved off with 3C PreScission protease overnight at 4 °C. The flow-through from the resins was collected and applied on a 6 ml Resource Q column (QIAGEN) to remove contaminants before purification by size exclusion chromatography using a Superdex 75 16/60 column (GE Healthcare) in the gel filtration buffer.

The securin$^{\Delta D box}$ mutant has the consensus residues at positions 1, 4 and 7 mutated to Ala, while the securin$^{D2}$ mutant has the entire native D box replaced with residues 60–80 of cyclin A2.

**APC/C complex formation.** Purified APC/C$^{\Delta Apc1-300s}$ (lacking Apc1 residues 307–395)[48] was incubated with purified Cdc20 and Cdk2-cyclinA2-Cks2 (either wild-type or mutant) at a molar ratio of 1:1.5:3 on ice before being purified on a Superose 6 10/300 column (GE Healthcare). The APC/C-substrate complexes were crosslinked with a water soluble crosslinker BS3 (bis[sulfosuccinimidyl] suberate) (Thermo Scientific). Crosslinking conditions were optimized across a range of temperature, crosslinker concentration and reaction time and assessed by 4–12% NuPAGE Bis-Tris gels as well as 4–16% NativePAGE Bis-Tris gels (Thermo Fischer). The optimal conditions for APC/C complexes were found to be at 1 mM BS3 on ice for 30 min. The crosslinking reaction was quenched by addition of 50 mM Tris-HCl pH 8.0 and incubation on ice for 30 min. Crosslinked APC/C complexes were purified by a Superose 6 Increase 3.2/300 column (GE Healthcare) using the Microakta system (GE Healthcare).

To prepare the complex APC/C$^{MCC}$-Cdk2-cyclinA2-Cks2, the APC/C was first phosphorylated by the kinases Cdk2-cyclinA2-Cks2 (truncated version without the N-terminal degrons of cyclin A2 (residues 174–432, lacking any degrons in cyclin A2)) and Plk1 (described in[48]). This was then purified on a 1 ml Resource Q column (QIAGEN) to remove kinases and ATP before addition of a 2x molar ratio of purified MCC. The APC/C$^{MCC}$ complex was purified on a Superose 6 10/300 column in the gel filtration buffer. Purified APC/C$^{MCC}$ was mixed with Cdk2-cyclinA2-Cks2 at a molar ratio of 1:33 and applied to a Superose 6 Increase 3.2/300 column to remove excess substrates and then used for grid preparation for cryo-EM.

**Binding assay of Cdk2-cyclinA2-Cks2 to APC/C$^{MCC}$.** Pre-formed and purified APC/C$^{MCC}$ complex at 1 mg ml$^{-1}$ was mixed with 33 times excess (molar ratio) of either wild-type Cdk2-cyclinA2-Cks2 or Cdk2-cyclinA2$^{\Delta ABBA}$-Cks2 and incubated on ice for 15 min. The mixture was purified on a Superose 6 Increase 3.2/300 column using the Microakta system (GE Healthcare) and the eluted peak fractions were analysed by SDS-PAGE on 4–12% NuPAGE Bis-Tris gels. Control runs of APC/C$^{MCC}$ alone and Cdk2-cyclinA2-Cks2 alone were performed with the same conditions.

**Cyclin A2 ABBA peptide pull-down assays.** Biotinylated cyclin A2 ABBA peptide (Designer BioScience) with the sequence GGANSKQPAFTIHVDEAE was dissolved in the gel filtration buffer to a concentration of 1 mM. The ABBA peptide was mixed with purified wild-type Cdc20, Cdc20$^{I235S\&Y279E}$ and Cdc20$^{I235S\&V295K}$ and incubated with 50 µl StrepTactin beads (QIAGEN) at 4 °C for 2 h in an eppendorf tube. The mixture was centrifuged at 2800 × g for 2 min at 4 °C in a tabletop centrifuge (Eppendorf Centrifuge 5418 R) to remove unbound proteins (in the supernatant) and washed three times with the gel filtration buffer before eluting with 50 µl 1x SDS-PAGE dye and boiling at 98 °C for 5 min. The boiled mixture was centrifuged for 1 min and the supernatant (elution) was carefully removed. Individual samples were loaded onto a 4–12% NuPAGE Bis-Tris SDS-PAGE gel for analysis.

**Ubiquitination assays.** The ubiquitination assay was performed with 60 nM recombinant human APC/C (for Cdh1) or phosphorylated APC/C (for Cdc20) or APC/C$^{MCC}$, 90 nM UBA1, 300 nM UbcH10, 300 nM Ube2S, 70 µM ubiquitin, 2 µM substrates (or titration of concentration), 5 mM ATP, 0.25 mg ml$^{-1}$ BSA, 15 µM CDK1/2 inhibitor iii (Enzo Life Sciences) and 30 nM Cdc20 or Cdh1 in a 10 µl reaction volume with 40 mM Hepes pH 8.0, 10 mM MgCl$_2$ and 0.6 mM DTT. The intracellular concentration of APC/C is reported as 80 nM[38], and that of Cdc20 is 100 nM[37,38], BubR1 90–130 nM and Mad2 (120 nM)[38], also references within ref. [54]. Cdk2 was estimated at 120–350 nM, whereas cyclin A2 and cyclin B1 were estimated at ~10–13 nM[39].

Reaction mixtures were incubated at room temperature for 30 min or various time points and terminated by adding SDS/PAGE loading dye. Cdk2 activity was inhibited by addition of 15 µM CDK1/2 inhibitor iii (Enzo Life Sciences), thereby preventing APC/C, coactivator, MCC and substrate phosphorylation. In the absence of the Cdk1/2 inhibitor the activity of APC/C$^{Cdc20}$ was greatly reduced (Supplementary Fig. 1a). Reactions were analysed by 4–12% NuPAGE Bis-Tris gels followed by western blotting with an antibody against the His-tag of ubiquitin to detect the His-tag of the ubiquitin-modified substrates (Clonetech, mouse monoclonal, 631212) and HRP-conjugated sheep anti-mouse antibody (GE Healthcare, NXA931V). Control gels showing the unmodified substrate for representative reactions are shown in Supplementary Fig. 2. Detection of cyclin A2 was performed with the rabbit monoclonal cyclin A2 antibody (Abcam, ab32386), detection of cyclin B1 with the rabbit monoclonal cyclin B1 antibody (Abcam,

ab32053), detection of securin with the rabbit monoclonal securin antibody (Invitrogen, 700791), detection of Apc4 with the rabbit monoclonal Apc4 antibody (Abcam, ab72149) and HRP-conjugated donkey anti-rabbit antibody (Thermo Fischer, SA1-200). Primary antibodies were used at a dilution of 1:1000 and secondary antibodies at a dilution of 1:5000.

The following peptides (Designer BioScience) were used in the ubiquitination assays: cyclin A2 6080 peptide (GLAQQQRPKTRRVAPLKDLPV) and Hsl1 D box peptide (SKRAALSDITNSSD). The peptides were dissolved at a concentration of 10 mM in the gel filtration buffer.

**Tissue culture and generation of stable cell lines**. HEK293 FlpIn-TRex cells (Invitrogen, cat. Number R78007) were cultured in DMEM (Gibco) supplemented with 10% tetracycline free FBS (PAN Biotech) at 37 °C and 5% $CO_2$.

For the stable integration of eGFP-cyclin A2 into the genome, all variants were cloned into the pcDNA5-FRT-TO vector (Invitrogen) with an N-terminal eGFP tag. HEK293 FlpIn-TREX cells were transfected with the pcDNA5-FRT-TO plasmids and pOG44, containing the flippase, using the HBS method. In short, cells were seeded the evening before transfection and the medium was exchanged the next morning. Both plasmids were mixed with 160 mM $CaCl_2$ and 2x HBS buffer (final concentrations: 137 mM NaCl, 5 mM KCl, 0.7 mM $Na_2HPO_4$, 7.5 mM D-Glucose, 21 mM HEPES) and added to the cells. The subsequent steps were performed according to the Invitrogen manual. Cells were selected using 100 ug mL$^{-1}$ Hygromycin B gold (InVivoGen). All stable cell lines were always kept under selection. Cyclin A2 detection in western blot was performed with the rabbit monoclonal cyclin A2 antibody (Abcam, ab32386).

**Live cell microscopy**. At a density of 75,000 cells per well, cells were seeded in an eight-well glass bottom microscopy slide (Ibidi) pre-coated with poly-L-lysine (Sigma) the evening before the experiment. To induce the expression of the eGFP-cyclin A2 variants, 0.25 ng mL$^{-1}$ doxycycline (Sigma) was added and the cells were synchronized in S-phase by the addition of 1 mM thymidine (Sigma). The next morning cells were released from thymidine by washing five times with fresh medium, while doxycycline was kept present. After 4–5 h, the medium was changed to $CO_2$-independent medium (Gibco) supplemented with 10% tetra-cycline free FBS (PAN Biotech) and GlutaMax (Gibco). The medium also contained 500 nM SiR-DNA (Chromotek) to visualise the DNA during live cell imaging and in some cases S-trityl-L-cysteine (STLC, Sigma) to induce a SAC arrest. After 1 hour of incubation the imaging was started using a SP8 confocal microscope (Leica) equipped with a heated environmental chamber, an argon laser, a 630 nm laser line and a ×40/1.1 numerical aperture water immersion lens. Two Z-stacks were defined with 5 μm spacing and the imaging was performed for 6 h (in the case of the SAC videos) or 14 h (in case of unperturbed mitosis). For the SAC videos the imaging was paused and a final concentration of 500 nM reversine (Sigma) was added to the cells. The positions were adjusted in their Z-stack to compensate for the additional medium (which took around 10 min) and the imaging was continued for one additional hour.

**Quantification of ubiquitination assay and fluorescence images**. Quantification of the ubiquitination assays were performed in ImageJ and plotted using Prism 8 (Graphpad). Fluorescence intensity of cells was measured using ImageJ (NIH) as described in ref. [55], with the exception that only a smaller circle inside the cell was chosen to measure the fluorescence intensity. It was ensured that the circle was always inside the cell and represented the overall behaviour of the eGFP signal. If this was not achievable the circle was adjusted or moved and the measurement of the effected time points was repeated. Data analysis and calculation of the eGFP-cyclin A2 degradation rate were performed using Excel (Microsoft) and graphs were created with Prism 7 and Prism 8 (Graphpad).

A two tailed Student's t-test was used to calculate the significance, except for data represented in Supplementary Fig. 3b, d, e, where a simple ANOVA test followed by Dunnett's multiple comparison test was used. One star indicates significance smaller than 0.05, two stars indicate significance smaller than 0.01, three and four stars indicate significance smaller than 0.001 and 0.0001 respectively. Statistical analysis was performed with Prism 8 (Graphpad). All experiments were repeated at least two times and all measurements were taken from independent samples. All values in the graphs are represented as mean ± s.d. (standard deviation). The number of cells analysed is given in the corresponding figure legends. The exact p-values are summarised in Supplementary Table 2.

**Cryo-electron microscopy**. For cryo-EM, 2 μl aliquots of freshly purified APC/C complexes at ~0.15 mg ml$^{-1}$ were applied onto Quantifoil R2/2 grids or R3.5/1 grids coated with a layer of continuous carbon film (~50 Å thick). Grids were treated with a 9:1 argon:oxygen plasma cleaner (Fischione Instruments Model 1070 Nano clean) for 20–40 s depending on the thickness of the carbon before use. The grids were incubated for 30 s at 4 °C and 100% humidity before blotting for 5 s and plunging into liquid ethane using a FEI Vitrobot III. The grids were loaded into a FEI Tecnai Polara or a FEI Titan Krios electron microscope operating at an acceleration voltage of 300 kV. Cryo-EM data for APC/C$^{\Delta\Delta Apc1-300s}$ in complex with Cdc20 and wild-type Cdk2-cyclinA2-Cks2 were collected at the Diamond Light Source on the FEI Titian Krios at a pixel size of 1.047 Å pixel$^{-1}$ using the Gatan K2

detector in electron counting mode. The rest of the cryo-EM data were collected on LMB microscopes, either on the FEI Tecnai Polara at a pixel size of 1.06 Å pixel$^{-1}$ using the FEI Falcon III direct electron detector in integration mode (APC/C$^{MCC}$-Cdk2-cyclinA2-Cks2) or on the FEI Titan Krios at a pixel size of 1.1 Å pixel$^{-1}$ using the Gatan K2 detector in electron counting mode (APC/C$^{\Delta\Delta Apc1-300s}$-Cdc20-Cdk2-cyclinA2$^{\Delta\Delta BBA}$-Cks2 and APC/C$^{MCC}$). Micrographs were recorded with a defocus range of −2.0 to −4.0 μm in integration mode and a defocus of −0.5 to −3.0 μm in electron counting mode. The exposure time for each micrograph was 1.2 s at a dose rate of 26 electrons Å$^{-2}$s$^{-1}$ for the Falcon III integration mode data collection and 36 movie frames were recorded for each micrograph, while 7 s exposure at a dose rate of 5 electrons pixel$^{-1}$s$^{-1}$ was used in the K2 electron counting mode and 28 movie frames were recorded for each micrograph.

**Image processing**. All movie frames were aligned by motioncor2[56] program before subsequent processing. First, the contrast transfer function (CTF) para-meters were calculated with Gctf[57]. Particles in 400 × 400 pixels were selected by automatic particle picking in RELION 3.0[58]. The following steps were performed to exclude bad particles from the dataset: (1) automatically picked particles in each micrograph were screened manually to remove ice contaminations[59]; (2) after particle sorting in RELION, particles with poor similarity to reference images were deleted; (3) two-dimensional classification was performed and particles in bad classes with poorly recognizable features were excluded. The remaining particles were refined using RELION three-dimensional (3D) refinement and divided into eight classes using 3D-classification in RELION with fine-angle search. During this process leftover bad particles were removed. Following three-dimensional-refinement of all good particles, fine-angle search three-dimensional classification was performed to separate the APC/C complex in different conformations. Beam-induced particle motion was corrected using Bayesian polishing in RELION 3.0. Particle subtraction was performed for the region of interest, either the substrate recognition site formed by Cdc20$^{WD40}$ and Apc10 or the MCC, using a soft mask (initial extension 3, soft edge 6) surrounding the region of interest followed by focused 3D-classification. Classes with improved coactivator or MCC density were 3D-refined in RELION individually. Multiple rounds of particle subtraction, focused 3D-classification and 3D-refinement were performed to separate different conformations. Final maps were sharpened and filtered in RELION Post-processing. Local resolution of the maps was estimated using RELION. All reso-lution estimations were based on the gold-standard Fourier Shell Correlation (FSC) calculations using the FSC = 0.143 criterion. A summary of all EM reconstructions obtained in this paper is listed in the Supplementary Table 1.

**Map visualization**. Figures were generated using Pymol (The PyMOL Molecular Graphics System, Version 2.0 Schrödinger, LLC.) and Chimera[60].

**Sequence alignment**. Sequence alignment was performed using Jalview[61].

## Data availability
EM maps are deposited in the Electron Microscopy Data Bank under accession codes: 4463 (APC/C$^{MCC}$-Cdk2-cyclinA2-Cks2 open conformation) [https://www.ebi.ac.uk/pdbe/entry/emdb-EMD-4463], 4464 (APC/C$^{MCC}$-Cdk2-cyclinA2-Cks2 closed conformation) [https://www.ebi.ac.uk/pdbe/entry/emdb-EMD-4464], 4465 (APC/C$^{\Delta\Delta Apc1-300s}$-Cdc20-Cdk2-cyclinA2-Cks2 D1 class) [https://www.ebi.ac.uk/pdbe/entry/emdb-EMD-4465], 4466 (APC/C$^{\Delta\Delta Apc1-300s}$-Cdc20-Cdk2-cyclinA2-Cks2 D2 class) [https://www.ebi.ac.uk/pdbe/entry/emdb-EMD-4466], and 4467 (APC/C$^{\Delta\Delta Apc1-300s}$-Cdc20-Cdk2-cyclinA2$^{\Delta D1}$-Cks2) [https://www.ebi.ac.uk/pdbe/entry/emdb-EMD-4467]. Protein coordinates are deposited in the Protein Data Bank under accession codes: 6Q6G (APC/C$^{\Delta\Delta Apc1-300s}$-Cdc20-Cdk2-cyclinA2-Cks2 D1 class) and 6Q6H (APC/C$^{\Delta\Delta Apc1-300s}$-Cdc20-Cdk2-cyclinA2-Cks2 D2 class). The source data underlying Figs. 1b, d, f, 2a, b, e, 3a, b, d, e, 4k, 5a, c, 6b–d, e and g and Supplementary Figs. 1a–g, i, j, 2, 4a, d, 6a, d–f and 6h are provided as a Source Data file. Cell lines that were generated for and used in this study are available upon request from the authors.

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

## Acknowledgements

This work was funded by MRC grant (MC_UP_1201/6) and CR-UK grant (C576/A14109) to D.B. PhD funding for S.Z. was from the Gates Cambridge Scholarship and Boehringer Ingelheim Fonds. We thank J. Pines for insightful discussion on the paper, C. Alfieri for discussion and critical reading of the manuscript and S. Aibara for advice on data processing. We are grateful to members of the Barford group for discussion; S. Chen and G. Cannone for EM facilities, J. Grimmet and T. Darling for computing, and J. Shi for tissue culture. The authors would like to thank Diamond Light Source for data collection time on Titan Krios M02 and M03 (EM17434-24), and the staff for assistance with data collection.

## Author contributions

S.Z. cloned and purified all proteins, performed all biochemical analysis, prepared grids, collected and analysed cryo-EM data and determined the three-dimensional reconstructions. T.T. performed live cell microscopy and in vivo data analysis. S.Z., T.T. and D.B. designed the project and experiments. S.Z. and D.B. wrote the manuscript with input from T.T.

## Additional information

**Competing interests:** The authors declare no competing interests.

**Peer Review Information:** *Nature Communications* thanks the anonymous reviewers for their contribution to the peer review of this work. Peer reviewer reports are available.

