## [Peer Review File · Nature Communications]

Reviewers' comments:

Reviewer #1 (Remarks to the Author):

This paper addresses an important problem in cell biology: the mechanisms by which the spindle assembly checkpoint (SAC) alters the substrate selectivity of the APC/C, the ubiquitin ligase that initiates chromosome segregation in anaphase. In the presence of chromosomes that are not correctly aligned on the spindle, the SAC triggers formation of a protein complex (the MCC) that binds the APC/C and inhibits its activity toward some substrates (securin, cyclin B) but not others (cyclin A2). Past work has provided numerous clues about the multiple degron sequences and other mechanisms that contribute to this selectivity. In the current paper, the authors use a blend of biochemistry, cell biology, and structural biology to provide interesting new insights into this problem. Most importantly, they identify a previously unnoticed APC/C degron on cyclin A2, and they show that this degron cooperates with previously known degrons to promote APC/C-mediated ubiquitination of cyclin A2 in the absence and presence of the MCC. A major strength of the paper, which distinguishes it from other work in the field, is that the conclusions are supported by structural studies that provide intriguing insights into underlying mechanisms.

Concerns:

1. In Figure 6a, b, size exclusion chromatography is used to support the authors' claim that the Cdk2-cyclin A-Cks2 complex binds to APC/C-MCC without displacing the MCC. This conclusion is not well supported by the protein gel analysis. The amount of kinase complex associated with APC/C-MCC is clearly substoichiometric, and the kinase complex is dissociating during the column run. Thus, displacement of MCC would be difficult to quantify by simply looking at the MCC gel bands in the complex or in other fractions. Ideally, it would be helpful to see the results from parallel runs of APC/C-MCC or MCC on this column. The authors should soften their language in their interpretation of these results.
2. The results in Figure 7c and d are disappointing. Much of the paper is devoted to the importance of the D2 degron, and the lack of a clear effect when transplanted to cyclin B1 is a rather unfortunate way to end the paper. Given that this negative result could be due to a variety of uninteresting explanations, it could be argued that these results could be removed, allowing the paper to end on the strong points raised in Fig 7a and b.

Minor points:

1. I was initially confused by the authors' use of ubiquitination assays in which the reaction products are analyzed by western blotting with anti-ubiquitin, rather than by the more conventional analysis of the substrate. It might be helpful to summarize the method briefly in the legends.
2. On page 4-5, the text states: "a peptide modelled on residues 60-80 potentially suppressed ubiquitination". In fact, inhibition seems to occur at tens or hundreds of micromolar peptide concentration, which I would not call potent. I would expect a good degron to have affinities in the 1-10 μ M range.
3. The cartoons in Fig 4g-I are not ideal. In the D1 mode, the Cdc20 shift is not indicated, and it looks like D2 is bound to Cdc20.
4. Fig 7b: the legend does not make it clear that we are looking at a speculative model of how cyclin A2 binds, based on the previous APC-MCC structure.

Reviewer #2 (Remarks to the Author):

In their manuscript, Zhang et al. aim to shed new light on the mechanisms that control the ubiquitination of cyclin A2 by the APC/C to promote its early degradation in prometaphase, overcoming the SAC-dependent suppression of the APC/C activity towards other targets that are degraded later in the cell cycle, like cyclin B1. The authors identified a new non-canonical D box (D2) in cyclin A2 and evaluated the role of this degron sequence and how does it act in concert with the other three previously described degrons in this protein (a canonical D box (D1), a KEN box and an ABBA motif) to facilitate its recognition by the APC/C and to modulate the activity of this ubiquitin ligase towards cyclin A2. Specifically, the results in their manuscript show that mutation of the D2 box in cyclin A2 results in significantly reduced levels of the APC/C-dependent ubiquitination of this cyclin, which indicates that the D2 sequence is critical for its degradation. Furthermore, the D2 box, together with the KEN box and ABBA motif, allow cyclin A2 to overcome the inhibition of the APC/C activity by the SAC and be ubiquitinated by the APC/C in its MCC-repressed state. Finally, Zhang et al. postulate that cyclin A2 can associate with APC/C^{Cdc20} in either a high activity binding mode, mediated by the D2 and KEN boxes, or a low activity mode via the ABBA motif and D1 and KEN boxes. Their results uncover interesting new aspects of the regulation of the activity of APC/C towards its different substrates and make the manuscript potentially suitable for its publication in Nature Communications. However, and although the experiments described by the authors are mostly well executed and presented, important controls are missing in some cases. On the other hand, there are some observations that necessitate clarification and conclusions that need to be strengthened with additional experimentation. Hence, I consider that the manuscript is still too preliminary in its present form and would require further experimental support to grant its final publication.

My main concerns about the results are the following:

- 1.- The text is often confusing, especially when referring to repression of the APC/C complex activity by the MCC. As a representative example, in the sentence: "Our results reveal that the resistance of cyclin A2 to MCC inhibition is conferred... by cooperative binding of four factors..." (page 10), one could mistakenly think that it is the MCC that is inhibited, and not that MCC inhibition of the APC/C is prevented. I would recommend to modify the manuscript, rephrasing the text in order to clarify all the instances where it could lead to this type of confusion.
- 2.- In most cases, the ubiquitination assays lack a negative control to ensure that the reaction is indeed APC/C-dependent, and therefore no ubiquitinated forms are observed in the absence of this ubiquitin ligase. Also, total levels of non-ubiquitinated Cyclin A2 or B1, and not only ubiquitinated forms starting from Ub1, should be displayed.
- 3.- Even though cyclin A2 is an early APC/C^{Cdc20} substrate, Supplementary Fig. 1a and 1b indicate that it is more efficiently and heavily ubiquitinated by APC/C^{Cdh1}. Could the authors comment about this observation?
- 4.- It would help to incorporate in all ubiquitination experiments some kind of quantification, similar to that shown in Fig. 7d, that helped with evaluating the extent to which the different substrates are modified by the APC/C.
- 5.- In Supplementary Fig. 1d it is stated that APC/C^{MCC} efficiently ubiquitinates cyclin A2 when is added at a concentration 33 times higher than that of the APC/C. How does this translate in terms of the intracellular concentration of the proteins? In general, it would be easier to interpret the results if the approximate relative intracellular concentration of Cyclin A2, APC/C, Cdc20, Cdh1 or MCC were provided.
- 6.- The authors postulate that the ABBA motif of cyclin A2 displaces the weaker second ABBA motif (A2) of BubR1, thereby associating with Cdc20. I wonder whether it would be possible to switch the ABBA motives of cyclin A2 and BubR1. If their model for how Cdk2-cyclinA2-Cks2 can be ubiquitinated by the MCC-repressed APC/C is correct, the prediction would be that, having a weaker ABBA motif, now cyclin A2 could not displace BubR1 and Cdk2-cyclinA2-Cks2 would be thus resistant to APC/C^{MCC} degradation.
- 7.- The authors should more extensively speculate about the possible physiological roles of the two binding modes of cyclin A2 with APC/C^{Cdc20}, since this observation could turn out to be just the result of an artificial conformation that is favored in the absence of the D2 box but

with no function whatsoever in the regulation of the activity of Cdk2-cyclinA2-Cks2.

8.- There is a general lack of statistical analyses (see Fig. 6d, 6f, 7d or Supplementary Fig. 2b, 2d, 2e), which are necessary to evaluate the significance of the differences shown between the different conditions that are compared.

Finally, some minor points are:

1.- In page 4, line 119, the initial sentence reads: "Previously findings..." [sic], instead of "Previous findings...".

2.- In page 10, lines 310-311, the sentence: "... how cyclin A2 overcomes MCC inhibition and the roles individual degrons in cyclin A2 play..." [sic] should be corrected ("...the roles that individual degrons...").

Reviewer #3 (Remarks to the Author):

Zhang, Tischer, and Barford present an interesting paper building on Barford lab expertise on structural biochemistry of APC/C, coupled with Tischer's in imaging in cell biology, to address the longstanding question of how Cyclin A2 is ubiquitylated/degraded during the spindle assembly checkpoint when other APC/C substrates are not. The authors take a systematic approach that leads to reporting a new, additional D-box "D2" sequence (previously recognized due to harbouring a Val replacement for the canonical Arg in the RxxLxD sequence). The authors show that this D2-box is crucial for Cyclin A2 recruitment to APC/C in some contexts. In addition to the identification of this Cyclin A2 degron, the paper has several major conclusions of broad significance:

- The authors show that two different degrons (here two different D-boxes, a KEN-box, an ABBA motif and presumably CKS protein binding to phosphorylated APC3 (although this is not experimentally addressed, it is implied) can be combined differently to achieve different substrate orientations on APC/C.

- Different degrons can contribute differently to substrate recruitment to APC/C activated by CDH1 and CDC20, and different rates of substrate degradation in cells.

- Cyclin A2's ability to bind APC/C MCC depends on its strong ABBA motif competing off a weak ABBA motif from MCC, which along with D-box binding allows binding to CDC20M and ubiquitylation in the context of an APC/C-CDC20-MCC complex.

- Cryo EM structures of APC/C-CDC20-Cyclin A2-CDK2-CKS2 complexes could be interpreted in light of the mutagenesis to explain potential arrangement of substrate degrons.

- Highlights of these structures include that CDC20 can be positioned slightly differently positions during ubiquitylation of different substrates.

Overall, these are significant and important contributions to our understanding of cell cycle regulation, the spindle assembly checkpoint, and of this fascinating molecular machine, APC/C. The experiments are high quality and thorough, and should be published in Nature Communications. While the authors have made a tremendous systematic effort, the system is complex, leading to some issues that should be addressed, at least through changes to the text, prior to publication.

1. The order of the presentation is a little confusing, because the authors first define the degrons and through mutations of most of them individually and together determine their roles, then determine some structures, then perform comprehensive analysis of the degron roles and then determine another structure. I recognize this order is probably first to define the D2-box, then see structures, then analyse the structures, but overall the presentation becomes somewhat confusing

to the extent that only the most intrepid reader interested in APC/C will make it through to the end. While the mitosis and APC/C field are big enough to make this a pretty broad group, the work is really nice and could be appreciated by an even broader audience if the order of presentation were improved. The authors can figure out for themselves what they prefer, but I might recommend putting all the biochemistry up-front. This suggests that multiple substrate orientations on APC/C may be possible. Then the structural work.

2. In reading the structural interpretations, the fact that the structures can actually only be interpreted in light of the mutational analysis should come before the analysis. For example, page 6 line 175 seems to me can only be concluded AFTER the subsequent paragraph starting at line 179. So I think something along the lines of that there were two major classes that are interpreted based on the mutations should be stated up-front before the description of the structures. This is especially true of the density for the complex with APC/C-CDC20-MCC, which to me looks in the figures identical to density for complexes without Cyclin A2-Cdk2-CKS2.

3. The authors should find a way to overlay closeups of the relevant regions of the new EM maps with each other and with previous maps to show the actual differences in the comparisons. It is important for the uneducated reader to be able to visualize whether or not there are obvious differences in the EM density, and the extremely major extent to which interpretation of the maps relies on knowledge obtained from the biochemistry. This is especially true for the complex with MCC, where I surmise that the only experimental evidence for presence of Cyclin A2-CDK2-CKS2 in the complex is the difference in the ratios of MCC open versus MCC closed conformations.

4. I may have missed this, but how can the authors exclude potential differences in phosphorylation accounting for the different conformations. I was not sure what construct was used to express CDK2. Many laboratories coexpress CDK2 with a CDK-activating enzyme, which would render the Cyclin A2-CDK2-CKS2 complex an active kinase. If active CDK2 was used, this needs to be clarified early in the text as a potential contributing factor to differences (although this reviewer does agree that the major differences in ubiquitylation pattern most likely result from degron differences). Even if non-CAK activated CDK2 was used, some weak activity of CDK2, in the context of high concentrations and a massive molar excess relative to the APC/C complexes could lead to some APC/C (or CDC20 or MCC) phosphorylation. Such phosphorylation would presumably depend on how Cyclin A2 is recruited to the different complexes (i.e., which spatial arrangements are possible – not just CKS2 recruitment to the APC3 loop in this case as is shown in the cartoon, but also recruitment of the Cyclin A2 degrons to CDC20 or CDC20M in the case of the MCC complex). This is especially true in light of the author's own data – that they cannot visualize a position for the globular kinase complex relative to the APC/C, implying that the flexibly tethered kinase active site could encounter potential substrates.

5. If the authors cannot rule out that the kinase has not changed the phosphorylation status of the APC/C-CDC20-MCC complex, do they have any additional data besides the change in ratio of open to closed conformation that the Cyclin A2-CDK2-CKS2 complex is actually associated with APC/C-CDC20-MCC in the particles visualized by cryo EM? If not, I don't think this precludes publication in Nature Communications with the data in-hand. However, the interpretation of the EM data becomes further muddied, and the authors need to simply state that they cannot absolutely determine if the density observed bound to MCC comes from the APC/C-CDC20-MCC itself or loosely tethered Cyclin A2-CDK2-CKS2, or whether the change in ratio of open/closed conformations comes from binding to Cyclin A2 degrons or from phosphorylation by this kinase.

6. Differences in phosphorylation may also contribute to the different degradation rates in cells. The cellular data are absolutely beautiful, but this potential contributing factor should just be stated. Although the rates of degradation of other substrates may be unaffected, there could be controls of phosphorylation at different stages of the cell cycle that correct for any effects of phosphorylation during the SAC.

We appreciate the positive comments and constructive and thoughtful suggestions of the reviewers. In response to their comments we have performed additional experiments that have strengthened our conclusions and improved the study. Reviewers' comments in red, our responses in black.

Reviewer 1.

Major points

1. In Figure 6a, b, size exclusion chromatography is used to support the authors' claim that the Cdk2-cyclin A-Cks2 complex binds to APC/C-MCC without displacing the MCC. This conclusion is not well supported by the protein gel analysis. The amount of kinase complex associated with APC/C-MCC is clearly substoichiometric, and the kinase complex is dissociating during the column run. Thus, displacement of MCC would be difficult to quantify by simply looking at the MCC gel bands in the complex or in other fractions. Ideally, it would be helpful to see the results from parallel runs of APC/C-MCC or MCC on this column. The authors should soften their language in their interpretation of these results.

We included an additional size exclusion chromatogram and associated SDS-PAGE gel for APC/C^{MCC} alone and compared this with the APC/C^{MCC}-Cdk2-cyclinA2-Cks2 complex (**Fig. 6a-d**). APC/C^{MCC} elutes slightly later than the APC/C^{MCC}-Cdk2-cyclinA2-Cks2 complex, consistent with Cdk2-cyclinA2-Cks2 associating with APC/C^{MCC}. The levels of MCC components associating with the APC/C do not vary between APC/C^{MCC} and the APC/C^{MCC}-Cdk2-cyclinA2-Cks2 complex, consistent with idea that Cdk2-cyclin A-Cks2 does not displace the MCC from APC/C^{MCC}. The text is revised (page 9).

2. The results in Figure 7c and d are disappointing. Much of the paper is devoted to the importance of the D2 degon, and the lack of a clear effect when transplanted to cyclin B1 is a rather unfortunate way to end the paper. Given that this negative result could be due to a variety of uninteresting explanations, it could be argued that these results could be removed, allowing the paper to end on the strong points raised in Fig 7a and b.

As suggested, we have removed the data relating to the substitution of the cyclin A2 D2 box for the cyclin B1 D box, but retain the result that adding the cyclin A2 KEN box and ABBA motif to cyclin B1 confers resistance to the SAC (**Fig. 7c, d**).

Minor points

1. I was initially confused by the authors' use of ubiquitination assays in which the reaction products are analyzed by western blotting with anti-ubiquitin, rather than by the more conventional analysis of the substrate. It might be helpful to summarize the method briefly in the legends.

We have clarified the assays in the legend of **Figure 1** and Methods (pages 15 and 16), and also as suggested by referee 2 included substrate loading controls for representative assays (**Supplementary Fig. 1e, 2**). The cyclin A2 and cyclin B1 antibodies do not detect ubiquitinated forms in our hands and therefore we blotted against the His-tag on ubiquitin.

2. On page 4-5, the text states: “a peptide modelled on residues 60-80 potentially suppressed ubiquitination”. In fact, inhibition seems to occur at tens or hundreds of micromolar peptide concentration, which I would not call potent. I would expect a good degron to have affinities in the 1-10 uM range.

Agreed, we have deleted ‘potently’.

3. The cartoons in Fig 4g-I are not ideal. In the D1 mode, the Cdc20 shift is not indicated, and it looks like D2 is bound to Cdc20.

Cartoons in Fig. 4g-i (now **Fig. 4h-j**) are modified to show the shift of Cdc20 position between **Fig. 4i** (the D2 box-binding mode) and **Fig. 4j** (the D1 box-binding mode) and the D2 box is displaced from Cdc20 in **Fig. 4j**.

4. Fig 7b: the legend does not make it clear that we are looking at a speculative model of how cyclin A2 binds, based on the previous APC-MCC structure.

The figure legend is modified to clarify that this is a speculative model of how cyclin A2 interacts with APC/C^{MCC}.

Reviewer 2.

Major points

1. The text is often confusing, especially when referring to repression of the APC/C complex activity by the MCC. As a representative example, in the sentence: “Our results reveal that the resistance of cyclin A2 to MCC inhibition is conferred by cooperative binding of four factors...” (page 10), one could mistakenly think that it is the MCC that is inhibited, and not that MCC inhibition of the APC/C is prevented. I would recommend to modify the manuscript, rephrasing the text in order to clarify all the instances where it could lead to this type of confusion.

Thank you for pointing out this confusion. We have altered the manuscript to clarify the text (pages 2, 3, 5, 9-11, 25, 27-29, 32).

2. In most cases, the ubiquitination assays lack a negative control to ensure that the reaction is indeed APC/C-dependent, and therefore no ubiquitinated forms are observed in the absence of this ubiquitin ligase. Also, total levels of non-ubiquitinated Cyclin A2 or B1, and not only ubiquitinated forms starting from Ub1, should be displayed.

We have performed additional ubiquitination assays to address this point. First, we have run negative controls (no E3 ligase) for the substrates cyclin A2, cyclin B1 and securin used in this study (**Supplementary Fig. 1e**). In the absence of the APC/C (as confirmed by anti-Apc4 antibody), no ubiquitination activity was detected. Second, as a control for substrate loading

levels when comparing the effects of cyclin A2 mutations on cyclin A2 ubiquitination, we show the level of unmodified as well as modified substrates by detecting with an anti-cyclin A2 antibody and an anti-His antibody (**Supplementary Fig. 2**). The assays were performed with APC/C^{Cdh1}, APC/C^{Cdc20} and APC/C^{MCC}. The cyclin A2 antibody does not detect ubiquitinated forms and therefore we blotted against the His-tag on ubiquitin to detect product formation. These new gels are representative of all ubiquitination reactions with cyclin A2 as a substrate.

3. Even though cyclin A2 is an early APC/C^{Cdc20} substrate, Supplementary Fig. 1a and 1b indicate that it is more efficiently and heavily ubiquitinated by APC/C^{Cdh1}. Could the authors comment about this observation?

We do not have a definitive explanation for this observation. In early mitosis, Cdh1 is inhibited by phosphorylation and APC/C^{Cdc20} is active to target cyclin A2 for degradation. One explanation might be that any residual low concentrations of cyclin A2 present during anaphase onwards are ubiquitinated by APC/C^{Cdh1}. This would require that APC/C^{Cdh1} has a higher affinity for cyclin A2 than APC/C^{Cdc20}.

4. It would help to incorporate in all ubiquitination experiments some kind of quantification, similar to that shown in Fig. 7d, that helped with evaluating the extent to which the different substrates are modified by the APC/C.

Quantification has been performed for assays using the same substrate with degron mutations (**Fig. 1b-g, Fig. 3b, c, Fig. 5a-d, Fig. 6e-h, Fig. 7c, d and Supplementary Fig. 1g, h**).

5. In Supplementary Fig. 1d it is stated that APC/C^{MCC} efficiently ubiquitinates cyclin A2 when is added at a concentration 33 times higher than that of the APC/C. How does this translate in terms of the intracellular concentration of the proteins? In general, it would be easier to interpret the results if the approximate relative intracellular concentration of Cyclin A2, APC/C, Cdc20, Cdh1 or MCC were provided.

From the literature we found the following estimations of intracellular concentrations for APC/C ~ 80 nM¹, Cdc20 ~ 100 nM^{1,2}, BubR1 90-130 nM and Mad2 120 nM¹ (also references within³). We found only one reference for Cdk2 and cyclin A2 and B1 concentrations⁴. Cdk2 was estimated at 120-350 nM, whereas cyclin A2 and cyclin B1 were estimated at ~ 10-13 nM⁴. We could not find a reference to the intracellular concentration of Cdh1. The intracellular concentrations of APC/C, Cdc20 and MCC are close to those used in our ubiquitination assays. The concentrations of substrates used are much higher, as judged based on the cyclin intracellular concentration of ref. ⁴. Now mentioned on pages 3 and 16.

6. The authors postulate that the ABBA motif of cyclin A2 displaces the weaker second ABBA motif (A2) of BubR1, thereby associating with Cdc20. I wonder whether it would be possible to switch the ABBA motives of cyclin A2 and BubR1. If their model for how Cdk2-cyclinA2-Cks2 can be ubiquitinated by the MCC-repressed APC/C is correct, the prediction would be that, having a weaker ABBA motif, now cyclin A2 could not displace BubR1 and Cdk2-cyclinA2-Cks2 would be thus resistant to APC/C^{MCC} degradation.

Our hypothesis that cyclin A2 possesses a stronger ABBA motif to displace the weaker second ABBA motif of BubR1 (ABBA³⁴⁰) is based on previous findings that the first two ABBA motifs of BubR1 are non-canonical ABBA motifs lacking the conserved Phe residue at P1 (ref. ⁵). Peptides of the first two BubR1 ABBA motifs could bind to Cdc20, yet with much lower

affinity than an ABBA peptide with the canonical sequence like that of cyclin A2 (ref. ⁵). It is a very interesting suggestion to swap the ABBA motif between cyclin A2 and BubR1, and it could be worthwhile to attempt in the future. Nonetheless, besides the ABBA motif, the KEN box and D2 box are also essential for cyclin A2 to overcome MCC-imposed inhibition. Swapping the ABBA motif of cyclin A2 to a weaker one may not have much effect as deletion of the entire ABBA motif did not make cyclin A2 resistant to APC/C^{MCC} ubiquitination.

7. The authors should more extensively speculate about the possible physiological roles of the two binding modes of cyclin A2 with APC/C^{Cdc20}, since this observation could turn out to be just the result of an artificial conformation that is favored in the absence of the D2 box but with no function whatsoever in the regulation of the activity of Cdk2-cyclinA2-Cks2.

The two binding modes of wild-type cyclin A2 to APC/C^{Cdc20} were observed by 3D-classification of the same cryo-EM dataset. Thus, the mutually exclusive binding of the D2 and D1 boxes to APC/C^{Cdc20} arise from the same dataset in the presence of all degrons.

We have also clarified this point by altering the text as follows: Two major 3D classes from the same dataset using wild-type Cdk2-cyclinA2-Cks2 as a substrate were characterized, corresponding to either the D1 box or D2 box bound to Cdc20^{WD40} (**Fig. 4b, d and Supplementary Fig. 5d, f**). Assignment of the two 3D classes to the D1 and D2 boxes was based on a cryo-EM structure of APC/C^{ΔApc1-300s} in complex with Cdc20 and Cdk2-cyclinA2^{ΔD1}-Cks2 lacking the D1 box, determined to 3.7 Å resolution (**Fig. 4c and Supplementary Fig. 4f**). In this structure, only a C-shaped density connecting Cdc20^{WD40} and Apc10 was observed (**Fig. 4c**), identifying this density as the D2 box (page 7).

We speculate on the possible physiological roles of the two binding modes (page 12). In principle, controlling which binding mode is adopted, possibly by restricting access to either the D1 or D2 box, would regulate the rate of cyclin A2 ubiquitination.

8. There is a general lack of statistical analyses (see Fig. 6d, 6f, 7d or Supplementary Fig. 2b, 2d, 2e), which are necessary to evaluate the significance of the differences shown between the different conditions that are compared.

Statistical analysis has been performed for **Fig. 1c, 1e, 1g, Fig. 3c, Fig. 5b, 5d, Fig. 6f, 6h, Fig. 7d and Supplementary Fig. 1h, 3b, 3d, 3e**. Details of the statistical analysis are described in the Methods section (pages 17-18) and the p-values are displayed in **Supplementary Table 2**.

Minor points:

1. In page 4, line 119, the initial sentence reads: “Previously findings...” [sic], instead of “Previous findings...”.

Thank you for spotting this. Corrected as suggested.

2. In page 10, lines 310-311, the sentence: “... how cyclin A2 overcomes MCC inhibition and the roles individual degrons in cyclin A2 play...” [sic] should be corrected (“...the roles that individual degrons...”).

Corrected as suggested.

Reviewer 3.

1. The order of the presentation is a little confusing, because the authors first define the degrons and through mutations of most of them individually and together determine their roles, then determine some structures, then perform comprehensive analysis of the degron roles and then determine another structure. I recognize this order is probably first to define the D2-box, then see structures, then analyse the structures, but overall the presentation becomes somewhat confusing to the extent that only the most intrepid reader interested in APC/C will make it through to the end. While the mitosis and APC/C field are big enough to make this a pretty broad group, the work is really nice and could be appreciated by an even broader audience if the order of presentation were improved. The authors can figure out for themselves what they prefer, but I might recommend putting all the biochemistry up-front. This suggests that multiple substrate orientations on APC/C may be possible.
Then the structural work.

We considered this helpful proposal very carefully, but eventually decided to retain the current layout with the change that we created a new sub-section to separate the cryo-EM work on the APC/C^{Cdc20}-Cdk2-cyclinA2-Cks2 complex from the biochemical work (Cryo-EM shows that cyclin A2 engages APC/C^{Cdc20} in two different binding modes). It is difficult to explain the results of the Δ D1 mutant of cyclin A2 without the context of the structural analysis that revealed the two binding modes of cyclin A2 to APC/C^{Cdc20} (i.e. one with the D1 box and the other with the D2 box bound to the D box receptor, with associated differences in activity).

2. In reading the structural interpretations, the fact that the structures can actually only be interpreted in light of the mutational analysis should come before the analysis. For example, page 6 line 175 seems to me can only be concluded AFTER the subsequent paragraph starting at line 179. So I think something along the lines of that there were two major classes that are interpreted based on the mutations should be stated up-front before the description of the structures. This is especially true of the density for the complex with APC/C-CDC20-MCC, which to me looks in the figures identical to density for complexes without Cyclin A2-Cdk2-CKS2.

Thank you for this suggestion. The text has been re-ordered.

3. The authors should find a way to overlay closeups of the relevant regions of the new EM maps with each other and with previous maps to show the actual differences in the comparisons. It is important for the uneducated reader to be able to visualize whether or not there are obvious differences in the EM density, and the extremely major extent to which interpretation of the maps relies on knowledge obtained from the biochemistry. This is especially true for the complex with MCC, where I surmise that the only experimental evidence for presence of Cyclin A2-CDK2-CKS2 in the complex is the difference in the ratios of MCC open versus MCC closed conformations.

A new figure has been generated that compares the D2 box of cyclin A2 with the canonical D box of Hsl1 published previously⁶ (**Fig. 4g**). Since the D2 box structure of wild-type cyclin A2 (D2 box-bound class) is identical to the D2 box structure of the Δ D1 mutant, a new figure was not included.

We have not included maps of super-imposed EM densities because this would be a little confusing. **Fig. 4b-d** shows EM density for the D1 and D2 boxes bound to the D-box receptor side by side: (i) density for the D2 box in the D2 box bound 3D class of wild-type cyclin A2 (**Fig. 4b**), (ii) density for the D2 box of the Δ D1 mutant (**Fig. 4c**) and (iii) density for the D1 box in the D1 box bound 3D class of wild-type cyclin A2 (**Fig. 4d**). Interpretation of the EM densities for assignment of the degrons is independent from the biochemical analysis as we determined the Δ D1 mutant structure for the assignment of the two D boxes.

For the APC/C^{MCC}-Cdk2-cyclinA2-Cks2 reconstruction, we did not observe any additional densities for Cdk2-cyclinA2-Cks2 and therefore could only speculate that the difference in the ratio of MCC open/closed states may arise from Cdk2-cyclinA2-Cks2 binding. The text has been modified as follows: An attempt to determine the cryo-EM structure of the APC/C^{MCC}-Cdk2-cyclinA2-Cks2 complex did not reveal additional densities accounting for Cdk2-cyclinA2-Cks2 (**Supplementary Fig. 4d-f, 6e and Supplementary Table 1**). Thus we cannot distinguish the degrons bound to the APC/C^{MCC} degron recognition sites are derived from the MCC or cyclin A2 (page 10).

4. I may have missed this, but how can the authors exclude potential differences in phosphorylation accounting for the different conformations. I was not sure what construct was used to express CDK2. Many laboratories coexpress CDK2 with a CDK-activating enzyme, which would render the Cyclin A2-CDK2-CKS2 complex an active kinase. If active CDK2 was used, this needs to be clarified early in the text as a potential contributing factor to differences (although this reviewer does agree that the major differences in ubiquitylation pattern most likely result from degron differences). Even if non-CAK activated CDK2 was used, some weak activity of CDK2, in the context of high concentrations and a massive molar excess relative to the APC/C complexes could lead to some APC/C (or CDC20 or MCC) phosphorylation. Such phosphorylation would presumably depend on how Cyclin A2 is recruited to the different complexes (i.e., which spatial arrangements are possible – not just CKS2 recruitment to the APC3 loop in this case as is shown in the cartoon, but also recruitment of the Cyclin A2 degrons to CDC20 or CDC20M in the case of the MCC complex). This is especially true in light of the author's own data – that they cannot visualize a position for the globular kinase complex relative to the APC/C, implying that the flexibly tethered kinase active site could encounter potential substrates.

We used Cdk2 phosphorylated at T160 that was produced in *E. coli* by coexpression of human GST-CDK2 and *S. cerevisiae* GST-Cak1 (Cak1 is also called Civ1) as described⁷ and D. Barford, J. Tucker, N.R. Brown and N. Hanlon, (unpublished data) (now clarified on page 13). The APC/C needs to be phosphorylated in order to be activated by Cdc20⁶. Therefore, when the APC/C is in complex with Cdc20 or the MCC, it has been *in vitro* phosphorylated by the kinases Cdk2-cyclinA2-Cks2 (truncated version without the N-terminal degrons of cyclin A2) and Plk1 and purified before complex formation or usage in the ubiquitination assays. It has now been clarified on pages 3 and 15 as well as in the figure legend of **Fig. 1**.

We performed ubiquitination reactions in the presence of 15 μ M CDK1/2 inhibitor iii (now mentioned on pages 3, 15 and 16). The CDK1/2 inhibitor (Enzo Life Sciences) is a potent ATP-competitive inhibitor towards Cdk1-cyclin B and Cdk2-cyclin A, with an IC₅₀ of 600 pM and 500 pM, respectively. Omitting the CDK1/2 inhibitor in the ubiquitination assay resulted in little ubiquitination. We included a new supplementary figure (**Supplementary Fig. 1a**) to demonstrate this. For the cryo-EM structures with Cdk2-cyclinA2-Cks2, both ATP and Mg were omitted from sample preparation, thereby preventing phosphorylation.

5. If the authors cannot rule out that the kinase has not changed the phosphorylation status of the APC/C-CDC20-MCC complex, do they have any additional data besides the change in ratio of open to closed conformation that the Cyclin A2-CDK2-CKS2 complex is actually associated with APC/C-CDC20-MCC in the particles visualized by cryo EM? If not, I don't think this precludes publication in Nature Communications with the data in-hand. However, the interpretation of the EM data becomes further muddled, and the authors need to simply state that they cannot absolutely determine if the density observed bound to MCC comes from the APC/C-CDC20-MCC itself or loosely tethered Cyclin A2-CDK2-CKS2, or whether the change in ratio of open/closed conformations comes from binding to Cyclin A2 degrons or from phosphorylation by this kinase.

We can exclude a difference in the ratios of open and closed APC/C^{MCC} arising due to phosphorylation by Cdk2-cyclinA2-Cks2 because both ATP and Mg required for phosphorylation were absent in the sample preparation procedure when Cdc20, MCC and Cdk2-cyclinA2-Cks2 were added.

To prepare the complex APC/C^{MCC}-Cdk2-cyclinA2-Cks2, the APC/C was first *in vitro* phosphorylated by the kinases Cdk2-cyclinA2-Cks2 (truncated version without the N-terminal degrons of cyclin A2) and Plk1. Following the phosphorylation reaction, the kinases, ATP and Mg were removed by anion-exchange chromatography. Purified phosphorylated APC/C was incubated with Cdc20 and the MCC to form APC/C^{MCC} that was then subjected to size exclusion chromatography. Once purified APC/C^{MCC} was obtained, wild-type Cdk2-cyclinA2-Cks2 was added to form a complex that was further purified by size exclusion chromatography and used for grid preparation for cryo-EM. The text is clarified on page 15. We have modified the text to be more circumspect that we cannot distinguish the degrons bound to the APC/C^{MCC} degron recognition sites are derived from the MCC or cyclin A2 (page 10).

6. Differences in phosphorylation may also contribute to the different degradation rates in cells. The cellular data are absolutely beautiful, but this potential contributing factor should just be stated. Although the rates of degradation of other substrates may be unaffected, there could be controls of phosphorylation at different stages of the cell cycle that correct for any effects of phosphorylation during the SAC.

We included the following sentence in the text on page 4 to address this point: The expression of mutant cyclin A2 proteins in these cell lines can be finely controlled by addition of doxycycline (**Supplementary Fig. 3a**) and a concentration was chosen to reflect the expression level of endogenous cyclin A2. While we cannot formally exclude that doubling the protein levels of cyclin A2 in cells by this approach influences the degree of phosphorylation present on Cdk/cyclin substrates, this effect seems to be negligible, because mitotic timing is not altered upon expression of different cyclin A2 mutants (except for non-degradable cyclin A2 mutants, see **Supplementary Fig. 3b**).

In addition, these degron mutations on cyclin A2 did not affect its binding to Cdk2, consistent with previous observation⁸. This further highlights our point that with regards to phosphorylation all cyclin A2 mutants behave the same as wild type cyclin A2 and that the differences in degradation rates we observed are not due to altered phosphorylation of the APC/C.

References

- 1 Tang, Z., Bharadwaj, R., Li, B. & Yu, H. Mad2-Independent inhibition of APC/Cdc20 by the mitotic checkpoint protein BubR1. *Developmental cell* **1**, 227-237 (2001).
- 2 Wang, Z., Shah, J. V., Berns, M. W. & Cleveland, D. W. In vivo quantitative studies of dynamic intracellular processes using fluorescence correlation spectroscopy. *Biophysical journal* **91**, 343-351, doi:10.1529/biophysj.105.077891 (2006).
- 3 Faesen, A. C. *et al.* Basis of catalytic assembly of the mitotic checkpoint complex. *Nature* **542**, 498-502, doi:10.1038/nature21384 (2017).
- 4 Arooz, T. *et al.* On the concentrations of cyclins and cyclin-dependent kinases in extracts of cultured human cells. *Biochemistry* **39**, 9494-9501 (2000).
- 5 Di Fiore, B., Wurzenberger, C., Davey, N. E. & Pines, J. The Mitotic Checkpoint Complex Requires an Evolutionary Conserved Cassette to Bind and Inhibit Active APC/C. *Molecular cell* **64**, 1144-1153, doi:10.1016/j.molcel.2016.11.006 (2016).
- 6 Zhang, S. *et al.* Molecular mechanism of APC/C activation by mitotic phosphorylation. *Nature* **533**, 260-264, doi:10.1038/nature17973 (2016).
- 7 Brown, N. R., Noble, M. E., Endicott, J. A. & Johnson, L. N. The structural basis for specificity of substrate and recruitment peptides for cyclin-dependent kinases. *Nature cell biology* **1**, 438-443, doi:10.1038/15674 (1999).
- 8 Di Fiore, B. *et al.* The ABBA motif binds APC/C activators and is shared by APC/C substrates and regulators. *Developmental cell* **32**, 358-372, doi:10.1016/j.devcel.2015.01.003 (2015).

REVIEWERS' COMMENTS:

Reviewer #1 (Remarks to the Author):

The authors have nicely addressed my previous comments, and I have no further concerns. This is an outstanding article and worthy of immediate publication.

Reviewer #2 (Remarks to the Author):

In the revised version of their original manuscript, the authors provide new data that reinforce the conclusions drawn. Zhang et al. have successfully clarified the main concerns raised and also included the requested additional controls for their experiments. Finally, and although the authors did not carry out some of the experiments suggested, they either provided a reasonable explanation or found an alternative approach to support their conclusions. Hence, I support final acceptance of their manuscript in Nature Communications. Nonetheless, I would like to make one final consideration about the updated version of the manuscript that, despite not being critical for acceptance, might be helpful to improve the final article. I do appreciate that the authors made a considerable effort to reorganize the text including all the indications of the reviewers and the new data generated, and that in order to do so it was probably necessary to cut down the number of words to maintain the length of the manuscript between the indicated guidelines. However, it is my impression that the beginning of the Results section was more straightforward to follow and also facilitated an easier understanding of the initial experiments in the original manuscript. Lastly, I would like to also point out a typo in the Methods section (line 541, Microsoft is misspelled).

Reviewer #3 (Remarks to the Author):

This reviewer feels that the the authors have addressed the questions of all referees. Overall, the manuscript beautifully illuminates exceedingly complicated regulation of the cell cycle. I recommend publication.

We are pleased that the reviewers are satisfied we have addressed their major concerns and we thank them for their suggestions and comments that have improved this paper. Our response in red.

Reviewer #1 (Remarks to the Author):

The authors have nicely addressed my previous comments, and I have no further concerns. This is an outstanding article and worthy of immediate publication.

We thank the reviewer for helpful comments and suggestions.

Reviewer #2 (Remarks to the Author):

In the revised version of their original manuscript, the authors provide new data that reinforce the conclusions drawn. Zhang et al. have successfully clarified the main concerns raised and also included the requested additional controls for their experiments. Finally, and although the authors did not carry out some of the experiments suggested, they either provided a reasonable explanation or found an alternative approach to support their conclusions. Hence, I support final acceptance of their manuscript in Nature Communications. Nonetheless, I would like to make one final consideration about the updated version of the manuscript that, despite not being critical for acceptance, might be helpful to improve the final article. I do appreciate that the authors made a considerable effort to reorganize the text including all the indications of the reviewers and the new data generated, and that in order to do so it was probably necessary to cut down the number of words to maintain the length of the manuscript between the indicated guidelines. However, it is my impression that the beginning of the Results section was more straightforward to follow and also facilitated an easier understanding of the initial experiments in the original manuscript. Lastly, I would like to also point out a typo in the Methods section (line 541, Microsoft is misspelled).

We thank the reviewer for helpful comments and suggestions.

1. We have re-structured paragraph 1 in the 'Results' section, moving the sentence on the intracellular concentrations of cyclin A2, APC/C etc. to the end of the paragraph. It is now more similar to the original.
2. Microsoft is now spelled correctly. Thank you for this.

Reviewer #3 (Remarks to the Author):

This reviewer feels that the authors have addressed the questions of all referees. Overall, the manuscript beautifully illuminates exceedingly complicated regulation of the cell cycle. I recommend publication.

We thank the reviewer for helpful comments and suggestions.